# Ultra-low loss quantum photonic circuits integrated with single quantum emitters

Ashish Chanana [1,2,3], Hugo Larocque[4], Renan Moreira[5], Jacques Carolan[4,10], Biswarup Guha[1,6], Emerson G. Melo [1,7], Vikas Anant[8], Jindong Song [9], Dirk Englund [4], Daniel J. Blumenthal [5], Kartik Srinivasan [1,6] & Marcelo Davanco [1] ✉

The scaling of many photonic quantum information processing systems is ultimately limited by the flux of quantum light throughout an integrated photonic circuit. Source brightness and waveguide loss set basic limits on the on-chip photon flux. While substantial progress has been made, separately, towards ultra-low loss chip-scale photonic circuits and high brightness single-photon sources, integration of these technologies has remained elusive. Here, we report the integration of a quantum emitter single-photon source with a wafer-scale, ultra-low loss silicon nitride photonic circuit. We demonstrate triggered and pure single-photon emission into a $Si_3N_4$ photonic circuit with $\approx 1\,dB/m$ propagation loss at a wavelength of $\approx 930\,nm$. We also observe resonance fluorescence in the strong drive regime, showing promise towards coherent control of quantum emitters. These results are a step forward towards scaled chip-integrated photonic quantum information systems in which storing, time-demultiplexing or buffering of deterministically generated single-photons is critical.

Advances have been made in photonic integrated circuit (PIC) technology based on wafer-scale ultra-low loss ($\approx 1\,dB/m$) waveguides (ULLWs). With propagation losses as low as $0.034\,dB/m$ at telecommunications wavelengths[1] and transparency from 405 nm through the infrared[2,3], the wafer-scale, CMOS compatible $Si_3N_4$ waveguide forms the basis of a versatile and promising integration platform. While focus has been on use of such technologies for classical applications, including coherent fiber communications[4], integrated microwave photonics[5], positioning and navigation[6] and atomic clocks[7], progress towards an ULLW integration platform for quantum applications has been limited. Overall, foundry-compatible quantum PIC platforms reported to date have featured waveguide propagation

losses of > 5 dB/m, as shown in Supplementary Table 1. Low photonic losses, including both waveguide propagation and insertion losses at on-chip components such as directional couplers, are central to meeting the scaling requirements for PICs that may be used to implement practical photonic quantum simulation[8], machine learning[9], and quantum computing[10], particularly with error correction[11]. Major loss contributions today that are detrimental to scaling include component insertion loss and waveguide interconnect loss between components like couplers, sources, and detectors. While insertion loss is a dominant factor in overall loss in quantum PICs, and must be reduced for producing throughputs comparable to those achievable in micro-optics circuits[12], PICs with ultra-low propagation

[1]Microsystems and Nanotechnology Division, Physical Measurement Laboratory, National Institute of Standards and Technology, Gaithersburg, MD, USA. [2]Institute for Research in Electronics and Applied Physics and Maryland NanoCenter, University of Maryland, College Park, MD, USA. [3]Theiss Research, La Jolla, CA, USA. [4]Department of Electrical Engineering and Computer Science, Massachusetts Institute of Technology, Cambridge, MA, USA. [5]Department of Electrical and Computer Engineering, University of California Santa Barbara, Santa Barbara, CA, USA. [6]Joint Quantum Institute, NIST/University of Maryland, College Park, MD, USA. [7]Materials Engineering Department, Lorena School of Engineering, University of São Paulo, Lorena, SP, Brazil. [8]Photon Spot, Inc., Monrovia, CA, USA. [9]Center for Opto-Electronic Materials and Devices, Korea Institute of Science and Technology, Seoul 02792, South Korea. [10]Present address: Wolfson Institute for Biomedical Research, University College London, London, UK. ✉e-mail: marcelo.davanco@nist.gov

losses will likely be critical for fault-tolerant photonic computing where photons must be 'stored' in delay lines[13], and also for quantum simulation schemes that rely on time-demultiplexing or buffering of single-photons, such as time-bin[14] or high-dimensional Gaussian Boson Sampling[15].

Bringing single-photon sources and ULLWs together on a single chip is critical for robustness, efficiency, performance, and compactness, especially for circuits that incorporate multiple independent sources. On-chip sources based on spontaneous four-wave mixing or spontaneous parametric down-conversion have been integrated within low-loss silicon-based and hybrid PIC platforms, with > 5 dB/m losses (see Supplementary Table 1). However, these sources exhibit a fundamental trade-off between the single-photon generation probability and purity, defined as the absence of multi-photon generation events, which limits the on-chip single-photon flux[16]. While multiplexing of multiple heralded sources can be employed to overcome this trade-off[17], it is challenging to simultaneously meet the phase-matching, high nonlinear coefficients and ultra-low losses with a single device layer on a chip, in particular since the requisite strong field confinement in high refractive index regions is detrimental to loss performance[18]. As an alternative, single quantum emitters do not suffer from the same purity versus brightness trade-off[19], and can produce pure streams of triggered single-photons at rates that are limited fundamentally by the cycling time between a ground and an excited state. Recently, integration of quantum emitter-based single-photon sources has been explored in homogeneous[20,21] or heterogeneous and hybrid PIC platforms[22,23] with waveguide losses in excess of 1 dB/cm. New solutions are needed that bring single quantum photon emitters onto ultra-low loss, ≤1 dB/m, waveguide technology in a wafer-scale CMOS compatible, scalable integration platform.

In this work, we report a significant advance towards this goal, in demonstrating the hybrid integration of ultra-low loss PICs and quantum emitter single-photon sources. Enabled by such capability, we envision the creation of quantum photonic circuits that may include not only the low-loss, large-scale, reconfigurable linear optical networks that implement quantum operations on chip, but also long on-chip delay lines for storing, time-demultiplexing or buffering of deterministically generated single-photons, as suggested in Supplementary Fig. 1.

Our PICs are based on a high aspect ratio, buried channel $Si_3N_4$ waveguide (WG) that is demonstrated here to achieve propagation losses of ≈1 dB/m at 930 nm. The quantum emitter that produces the single photons are single InAs quantum dots (QDs) embedded in GaAs nanophotonic geometries that utilize a tapered mode-transformer to efficiently couple to the $Si_3N_4$ ultra-low loss waveguide structures[24,25].

We report the demonstration of triggered emission of QD single-photons into ULLWs, with $g^{(2)}(0) < 0.1$, indicating high single-photon Fock-state purity. We also report the observation of waveguide-coupled single dot resonance fluorescence in the strong drive regime, evidenced by the appearance of the Mollow triplet in the QD emission spectrum. Such a feature is a signature of dressed states emerging from the coupling of a two-level system to a strong coherent excitation field[26,27], which is not only of scientific relevance, but which also offers prospects for the development of sources of single correlated photon pairs or photon bundles, which may find applications in e.g., multi-photon spectroscopy[28,29] or quantum communications[30].

## Results

### Device description and fabrication

Figure 1 shows a schematic of our hybrid integration platform. The ULLWs consist of a high-aspect ratio $Si_3N_4$ core, with a thickness of 40 nm and width of 2 μm, buried under 1 μm $SiO_2$ upper cladding layer. The top cladding thickness is chosen to ensure a weakly confined single transverse-electric (TE) guided mode with low propagation losses in the 900 nm wavelength band[31]. The on-chip single-photon source consists of a straight GaAs nanowaveguide with embedded InAs self-assembled QDs followed by an adiabatic mode transformer, a geometry that has been shown to allow efficient coupling of QD emission directly into air-clad $Si_3N_4$ ridge waveguides[25,32]. Opposite to the adiabatic taper, a one-dimensional photonic crystal back-reflector designed for high reflectivity above 900 nm is introduced to allow unidirectional emission into the $Si_3N_4$ waveguide. To ensure evanescent coupling between the GaAs and $Si_3N_4$ layers using the mode transformer, the QD-containing GaAs device is placed in direct contact

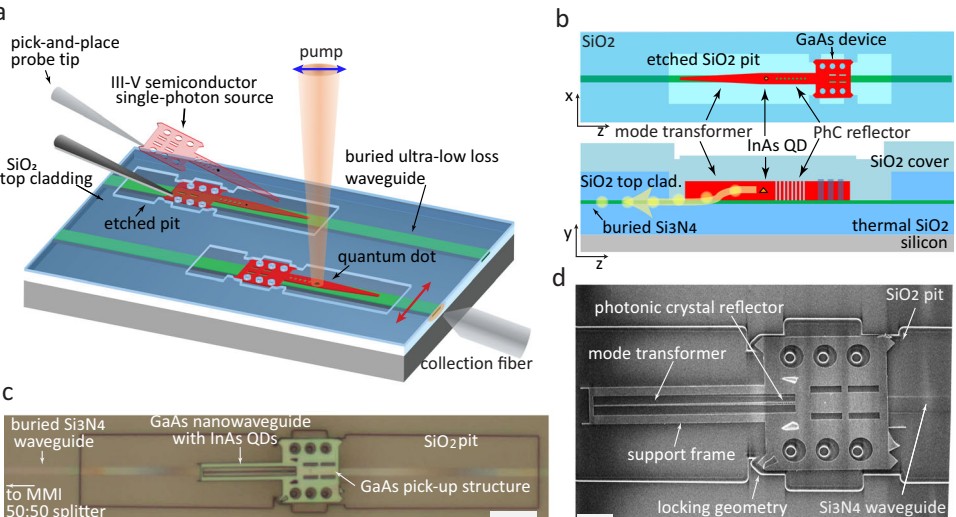

**Fig. 1 | Integration of single photon source to ultra low-loss waveguide.**
**a** Schematic of pick-and-place hybrid integration of a GaAs nanophotonic device containing InAs quantum dots (QDs) onto an ultra-low loss $Si_3N_4$ waveguide (ULLW). Tungsten probes were used to place and align the GaAs device to the etched pit and the buried ULLW. Control of the pump beam polarization (indicated by the blue arrow) allows resonant QD excitation with minimal pump scattering into the ULLW, allowing observation of resonance fluorescence coupled to the transverse-electric (TE) polarized mode (represented by the red arrow). **b** Top-view and cross-sectional schematic of hybrid device geometry. **c** Optical micrograph of a GaAs/InAs QD single-photon source assembled on a $Si_3N_4$ ultra-low loss waveguide, leading to a 50:50 multimode interference coupler (MMI) power splitter (not shown). The image was taken prior to the top $SiO_2$ cladding deposition. Scale bar: 10 μm **d** Scanning electron micrograph of the device prior to deposition of the $SiO_2$ top cladding. Scale bar: 4 μm.

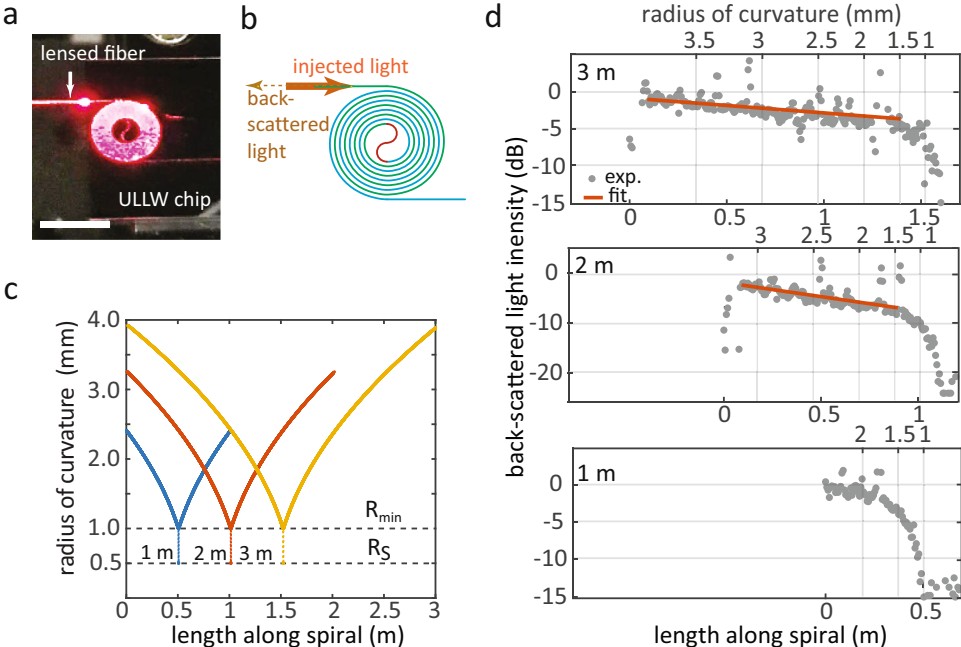

**Fig. 2 | Characertization of losses in ultra low-loss waveguide. a** Photograph of an ULLW spiral with 1 m length under test. Scale bar: 5 mm. **b** Schematic of the Archimedean spirals used for loss measurement, composed of inward (green) and outward (blue) spirals, connected by an S-bend (red). In the measurement, laser light is injected into the spiral, and guided back-scattered photons originating along the spiral, collected from the spiral input, are detected in time-domain with a resolution of ≈ 200 ps. **c** Radius of curvature (RoC) as a function of length for the measured 1 m, 2 m, and 3 m spirals. $R_{min}$ and $R_S$ respectively mark the minimum spiral and S-bend radii. We note the large RoC discontinuity at the S-bend. **d** Back-scattered light intensity as a function of propagation length and RoC along the 1 m, 2 m, and 3 m spirals, relative to the intensity at the start of each spiral (dots: data; red lines: fits). In each panel, the top and bottom horizontal axes are, respectively, the RoC and length along the corresponding spiral. The spiral length uncertainty is < 1 mm, as described in the Supplementary Note 3.

with the top of the $Si_3N_4$ guide. This is accomplished by first etching a pocket into the 1 μm top $SiO_2$ upper cladding of the ULLW, down to its $Si_3N_4$ core, and then placing the GaAs device into the pocket, as seen in Fig. 1a. In the following step, the placed GaAs device is covered with a 1 μm thick $SiO_2$ cladding layer, as shown in Fig. 1b, c. It is worth noting that portions of the $Si_3N_4$ ULLW that are distant from the placed GaAs device are completely unaffected by our processing, since the top $SiO_2$ cladding is preserved everywhere. Finite difference time-domain (FDTD) simulations predicted that the fabricated geometries could yield a maximum theoretical single-photon coupling efficiency $\eta_{QD\text{-}ULLW} \approx 0.31$ into the $Si_3N_4$ waveguide. Sections in the main text and Supplementary Notes 7 and 8 discuss concrete alternative geometries that have the potential to achieve $\eta_{QD\text{-}ULLW} > 0.8$.

The hybrid device fabrication is described in the Methods. Figure 1c shows an optical microscope image of an assembled GaAs nanowaveguide placed above a buried $Si_3N_4$ ULLW leading into a multi mode interference (MMI) 50:50 splitter. The outline of the etched $SiO_2$ cladding corresponding to the GaAs device placement pit is indicated in the figure. The nanowaveguide geometry, which hosts the quantum dot single-photon emitter, is surrounded by a frame created for mechanical alignment and structural support, and is connected to a pick-up pad that is used for transferring it onto the $Si_3N_4$ chip. As highlighted in the scanning electron micrograph (SEM) of Fig. 1d, the GaAs device geometry has auxiliary locking features complementary to those of the etched placement pockets, to facilitate alignment. A misalignment between the GaAs device and the $Si_3N_4$ waveguide of < 340 nm, as well as a tilt angle of < 0.9° can be inferred from Fig. 1d.

**Ultra-low loss waveguide characterization**

To estimate the propagation losses, guides with nominal lengths of 1 m, 2 m, and 3 m, implemented as Archimedean spirals[33], were fabricated and characterized by a single-photon optical time-domain

reflectometry (SP-OTDR) technique[34]. In this technique, short laser pulses, at a center wavelength of ≈ 930 nm, in a periodic stream are launched into the ULLW, and photons originating from optical back-scatter along the waveguide are collected and routed towards a single-photon detector. A time-correlator is then used to create a time-trace of back-scattered photon arrival times with respect to a reference clock, and the arrival time can be converted into a distance along the guide. The evolution of the back-scattered light intensity with arrival time provides a direct measure of the signal attenuation along the guide. The experimental setup and details about the measurements and time-to-length conversion are provided in Supplementary Notes 2 and 3. It is worth noting that while such a method has been employed in the past for characterizing fiber optic links[34], here we show that it may be used for characterizing on-chip ULLWs.

As shown in Fig. 2c, the Archimedean spirals were designed with a radius of curvature (RoC) that varied continuously going inwards, from a maximum value $R_{max}$—which depended on the total length—to a minimum $R_{min} = 1000$ μm near the center. The inward spiral was followed by an S-bend with $R_S = 500$ μm, which transitioned to the outward spiral towards the waveguide output. Time-domain reflectivity traces for the three spirals are shown in Fig. 2d, as a function of spiral length and RoC. All reflectivity curves are approximately linear (in log scale) up to about half of the total spiral lengths. Approximately at the S-bends, the signals drop precipitously. Transmission spectra (not shown) of waveguide-coupled microring resonators with radius $R = 500$ μm on the same chip did not reveal any resonances, indicating that the signal drop is due to large bend losses at the S-bends. It is also likely that the sharp RoC transition between the spiral and S-bend cause further signal loss. To estimate propagation losses in straight ULLWs (bent WGs are not subsequently used in QD integration), linear fits to the OTDR traces were used[33]. The fits were performed for $z$ values from the beginning of the

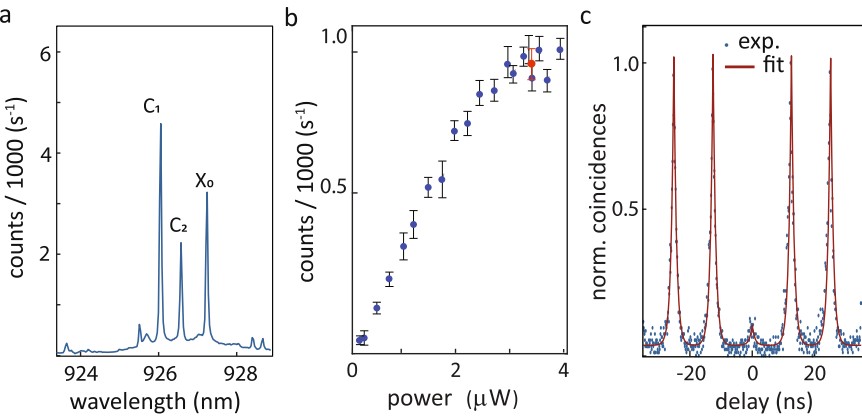

**Fig. 3 | Single photon emission measured via ultra low-loss waveguide.**
**a** Quantum dot photoluminescence (PL) spectrum from a hybrid device pumped non-resonantly at 841.5 nm, showing three transitions from the same QD. **b** PL intensity for $X_0$ as a function of input power. Red dot: pump level for measurements in **c**. The uncertainties represent 95% confidence intervals computed from a Lorentzian fit to the QD emission line intensity. **c** Second-order correlation for the $X_0$ line pumped at saturation, showing triggered single photon emission with fitted $g^{(2)}(0) = (0.07 \pm 0.03)$ at zero delay. All uncertainties reported are 95% fit confidence intervals, corresponding to two standard deviations.

inward spiral to 1 cm before the start of the S-bend, to avoid the abrupt RoC discontinuity. Linear losses for 3 m and 2 m spirals were found to be, respectively, $(1.0 \pm 0.4)$ dB/m and $(2.8 \pm 0.6)$ dB/m. Fits to the 1 m spiral trace did not yield reliable parameters, primarily due to the short extent of the available data.

## Triggered single-photon emission

We next demonstrate triggered single-photon emission from a single QD into a ULLW and characterize its spectral properties and photon statistics at temperatures < 10 K. Figure 3a shows the micro-photoluminescence ($\mu$PL) spectrum obtained for the device in Fig. 1c, pumped from free space with a continuous-wave laser at 841.5 nm, and collected from the ULLW (details in the Methods). The emission lines at 927.21 nm, 926.57 nm, and 926.02 nm (labeled as $X_0$, $C_2$, and $C_1$, respectively) were found to be from a single QD via photon-counting cross-correlation measurement. Characterization of the three emission lines is provided in Supplementary Note 4. To determine the purity of single photon emission, the second-order intensity correlation $g^{(2)}(\tau)$ line was measured in a Hanbury-Brown and Twiss setup. Figure 3c shows the normalized photon detection coincidences, where a fitted $g^{(2)}(0) = 0.07 \pm 0.02$ and decay parameter of $(0.85 \pm 0.02)$ ns was obtained, close to the radiative rate measured to be, $\tau_1 = (0.86 \pm 0.01)$ ns. This shows triggered high-purity, single-photon emission from the QD collected in the ULLW.

The single-photon count rates produced by the QD pumped into saturation were compared to the 80 MHz pulsed laser repetition rate to yield a measure of the QD-to-ULLW coupling efficiency $\eta_{\text{QD-ULLW}}$. Assuming 100% quantum efficiency for the $X_0$ line and discounting all photon losses along the optical path from the $Si_3N_4$ ULLW to the employed superconducting nanowire single-photon detector (SNSPD), we estimate $4\% \leq \eta_{\text{QD-ULLW}} \leq 7\%$. As detailed in Supplementary Note 5, the detector efficiency was $\approx 71\%$ and the system efficiency was $\approx 11\%$. Finite difference time-domain (FDTD) simulations of electric dipoles emitting in a hybrid geometry that approximated the fabricated and tested one indicate that $\eta_{\text{QD-ULLW}} < 31\%$ could in principle be achieved. As detailed in Supplementary Note 5, the discrepancy between experimental and simulated efficiencies is likely primarily due to sub-optimal QD position and dipole moment orientation inside the GaAs nanowaveguide, though contributions from the misalignment between the latter and the underlying ULLW (evident in Fig. 1d) and other geometrical imperfections were potentially significant. Potential steps to improve the coupling efficiency are expanded in the Discussion.

## Resonance fluorescence

An additional necessary characteristic for on-chip single-photon sources is high single-photon indistinguishably, which requires the benchmark $T_2 = 2T_1$ for the quantum emitter coherence time $T_2$, where $T_1$ is the radiative lifetime. Non-resonant excitation of the QD results in an excess of electrons and holes in the host semiconductor and leads to a fluctuating charge environment that inevitably leads to single photons with $T_2 \ll T_1$. Resonant QD excitation, on the other hand, has been shown to minimize decoherence, allowing the radiative limit to be approached, by avoiding excess environmental charge fluctuations[35]. An inherent challenge of such a scheme, however, is to sufficiently suppress a pump beam that is resonant with the quantum emitter fluorescence. In free-space-coupled systems, suppression is typically achieved through polarization filtering of the pump before detection[36], though excitation with an orthogonally directed free-space beam[37] or waveguide[38] has also been used, and a bi-chromatic pumping scheme has also been recently explored[39]. In PICs featuring direct quantum dot resonant illumination with a free-space beam, off-chip polarization filtering before detection has been employed[40,41], as well as temporal detection gating of on-chip superconducting nanowire superconducting detectors (SNSPDs)[42]. In our device and experimental configuration, we observed the resonance fluorescence spectrum collected directly into the ULLW, without polarization filtering or temporal gating. We measured an extinction ratio of > 25 dB using resonant laser excitation by controlling the polarization of the incident laser alone. This was made possible due to high spatial mode filtering provided by the high aspect ratio ULLW, which only supports a TE mode, so that the polarization orthogonal to the one supported by the waveguide is highly suppressed. We note that resonance fluorescence has also been observed without polarization filtering in AlN circuits with integrated Ge-vacancy quantum emitters in diamond[43], and control of pump polarization alone was sufficient to allow observation of waveguide-coupled resonance fluorescence with on-chip SNSPDs[44].

The resonance fluorescence spectrum of a two-level system varies significantly with excitation intensity. At excitation powers significantly below the saturation level, elastic resonant Rayleigh scattering dominates the observed spectrum, featuring an apparent linewidth narrower than the emitter's radiative limit. While observation of antibunching of such signal has initially been reported[45], its statistics has recently been shown to change significantly upon narrow spectral filtering[46,47], a behavior that has been explained as interference between coherent scattering and weak incoherent

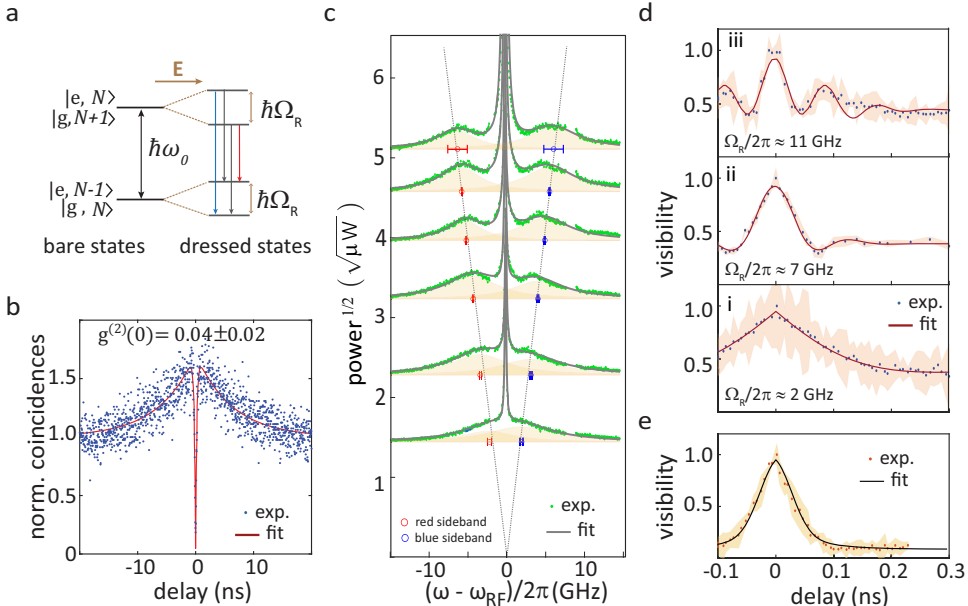

**Fig. 4 | Resonance fluorescence and coherent control of quantum dot. a** Energy levels of a two-level system (TLS) driven by a resonant coherent optical driving field. Electric dipole interaction with the driving field (**E**) splits the TLS 'bare' states into two 'dressed' states separated by $\hbar\Omega_R$, producing three emission peaks, represented by different colors. The labels e, g and $N$ respectively correspond to the TLS ground and excited states, and the coherent field average photon number. **b** Measured second-order correlation and fit (red line) for the QD emission under resonant excitation. **c** Strong-drive resonance fluorescence spectra (green dots) for increasing excitation powers and fits (gray lines). Gaussians used to fit the side-peaks are plotted as shaded areas. Red and blue Mollow side-peak positions from the fits are plotted in red and blue symbols, respectively. Linear fits to the side-peak energies with respect to the square-root of the excitation power are plotted as dotted gray lines. **d** Interferometric fringe visibility in Fourier transform spectroscopy (see Methods) as a function of time delay for QD emission. Panels (i) to (iii) are for resonance fluorescence at different excitation powers. **e** Fringe visibility for quasi-resonant (p-shell) pumping at 877.5 nm. In **d**, **e**, shaded areas indicate measurement uncertainties. All reported uncertainties correspond to 95% fit confidence intervals, corresponding to two standard deviations.

emission[46]. At high excitation power, the spectrum features a central resonant peak and two symmetric side-resonances, forming the so-called Mollow triplet[26,48]. Waveguide-coupled resonance fluorescence from single quantum emitters has previously been demonstrated in various single-material[40,41,49,50] and hybrid[43,51,52] PIC platforms. In contrast with all this prior work, below we report observation of the Mollow triplet in waveguide-coupled emission, from the same device as measured in the previous Section. The origin of the triplet can be understood from the schematic in Fig. 4a. Two bare states of the quantum dot-field system are split by electric dipole interaction with a strong excitation field, forming a quartet of dressed states. The doubly-degenerate transitions at the resonant energy and the blue- and red-shifted transitions compose the Mollow triplet. The side-peak splitting is given by the Rabi frequency, $\Omega_R$, which is proportional to the electric field amplitude.

To observe resonance fluorescence from our device, a free-space laser beam tuned to the $X_0$ transition in Fig. 3b was used. As detailed in the Methods, control of the pump beam polarization was used to minimize scatter into the ULLW, and a weak non-resonant co-pump was used to gate the resonant emission[53]. Resonantly driven single-photon emission was first verified via a second-order photon correlation measurement of the resonance fluorescence spectrum measured at excitation power of 7.7 μW. The data, shown in Fig. 4b, displays a clear anti-bunching dip, with a fitted $g^{(2)}(0) = 0.04 \pm 0.02$, without deconvolution, indicating nearly pure single-photon emission. A bunching peak at ≈ 3 ns, however, indicates flickering due to dark state shelving[54] or spectral diffusion[41], with a time-scale of ≈ 6.4 ns. Such behavior is likely due to a fluctuating charge environment surrounding the QD, which is ameliorated, though not completely suppressed, by the non-resonant co-pump[55]. We note also that the $X_0$ transition radiative lifetime $T_1$ was measured to be $T_1 = (0.63 \pm 0.01)$ ns, as shown in Supplementary Note 9, a

comparable value of $T_1$ was obtained from fit of $g^{(2)}$. The value is slightly shorter than previously measured under non-resonant excitation. Such discrepancy is likely due to a slower QD excitation dynamics in the latter case, leading to broadened lifetime traces[56].

Figure 4c shows high resolution resonance fluorescence emission spectra, obtained with a scanning Fabry-Perot interferometer (SFPI), for varying pump powers (details in Methods). The spectra display a sharp Lorentzian central peak and two side-peaks, spaced from the latter by an energy that varies linearly with the excitation field amplitude (square-root of the power), a signature of the Mollow triplet. The sharp central peak includes the elastic contribution of the Mollow spectrum, and scattered resonant pump light. The side-peaks show a slight asymmetry in amplitude and width, which suggests some detuning between the laser and the transition[57], and spectral diffusion, at time-scales $> T_1$[48]. Indeed, as shown in Supplementary Fig. 10, a model that takes into account QD spectral diffusion[48] is able to fit the data, yielding $T_2 < 100$ ps. To confirm and better estimate $T_2$, we use Fourier Transform spectroscopy[58]. Here, the resonant QD emission was fed into a variable-delay Mach-Zehnder interferometer, and output interference fringe amplitudes were recorded as a function of time-delay. The resulting traces, shown in Fig. 4d, are proportional to the first-order correlation function of the QD light[58], and were fitted to a model[59] that yielded the coherence time $T_2$, as well as the Rabi frequency $\Omega_R$ (see Methods and the Supplementary Note 10 for details). A reference visibility trace, obtained for non-resonant pumping, is shown in Fig. 4e. The trace is fitted with a weighted sum of a Gaussian and a two-sided exponential, where the Gaussian component indicates spectral diffusion, and yields $T_2 = (0.053 \pm 0.003)$ ns. Panels i to iii in Fig. 4d are visibility traces for resonance fluorescence for varying excitation powers, as indicated by the Rabi frequencies. It is worth noting that at the higher powers Rabi oscillations are visible, which

are reasonably well reproduced by the model[59]. The corresponding coherence times for panels i, ii and iii are $T_2 = (0.10 \pm 0.1)$ ns, $T_2 = (0.07 \pm 0.01)$ ns and $T_2 = (0.09 \pm 0.01)$ ns, longer than for the non-resonant excitation values. The coherence dynamics at high powers are a better fit with a Gaussian decay, while at lower powers coherence decays exponentially, which indicates the prevalence of spectral diffusion at high powers[58].

## Discussion

Our work demonstrates integration of a quantum emitter single-photon source onto photonic integrated circuits with waveguide losses of ≈1 dB/m. In contrast, losses in excess of 100 dB/m have to date been reported for photonic circuits with on-chip quantum emitter sources, and of at least 5 dB/m for foundry-compatible integrated quantum photonic circuits overall (see Supplementary Table 1). We next outline and discuss improvements to achieve the full potential of our integration platform.

Regarding the relatively low single-photon coupling efficiency into the ULLWs demonstrated here, the main contributing factors include a sub-optimal nanophotonic design and quantum dot positioning and, principally, dipole moment orientation within the GaAs device. While various techniques have been developed to solve the latter issues[32,56] the implemented photonic design featured two factors that fundamentally lead to lower efficiencies. First, the choice of a waveguide geometry imposes a limit on the QD coupling to guided, as opposed to radiative, waves[25]. Indeed, in Supplementary Note 6, a maximum $\beta \approx 88\%$ QD coupling efficiency was predicted into the straight GaAs waveguide section of our fabricated light sources. At the same time, the GaAs mode transformer leading to the $Si_3N_4$ waveguide, featuring an unoptimized linear width taper, was predicted to have only $\eta_{MT} \approx 35\%$ efficiency. Overall, a maximum source efficiency of $\beta \cdot \eta_{MT} \approx 31\%$ could be expected from the implemented geometries. As exemplified in Supplementary Note 7, however, properly optimized adiabatic mode transformers may be designed to be considerably more efficient ($\eta_{MT} > 93\%$), comparable to that achieved in non-ULLW platforms[22,23]. Improved QD coupling efficiencies $\beta$ into the straight GaAs waveguide section may also be obtained via the implementation of low-Q cavities, as also shown in Supplementary Note 8, which would lead to an improved overall $\eta_{QD\text{-}ULLW}$. Evanescently coupled microcavities are another viable, narrow-band alternative towards achieving higher overall coupling efficiencies[60] and are the subject of future work. An advantage of cavity-based approaches is that a high Purcell radiative rate enhancement, achieved through coupling to the resonant mode, can bring the quantum emitter's lifetime $T_1$ closer to the radiative limit $T_2 = 2T_1$, given a coherence time $T_2$ that is sufficiently unaffected by nanofabrication, thereby improving indistinguishability[56,61]. On the other hand, a single quantum dot exhibits various excitonic transitions over a relatively wide spectral range, which may be used for desirable functionalities beyond triggered single-photon emission. For instance, polarization-entangled photon pairs may be generated from the biexciton-exciton cascade[62], where the two states are typically split by ≈1 nm. These entangled photon states, when captured into an integrated photonic circuit-for instance via two TE modes of a multimode GaAs waveguide[63], could present interesting opportunities for quantum information processing on a chip. Importantly, all of the suggested options for improving the source efficiency would only involve modifications to the GaAs device layer, whereas the $Si_3N_4$ ultra-low loss portions of the circuit would remain unaffected.

Regarding collection of resonance fluorescence with higher pump suppression, fine control of the QD orientation will likely be necessary. Control of the resonant pump polarization was shown here to effectively minimize scatter into the ULLW. Keeping in mind that only the QD dipole moment component that is transverse to the ULLW couples to it, the QD must be oriented such that the (optimally polarized) pump maximizes resonant QD emission into the ULLW. The QD must have a sufficiently large dipole moment component along the pump polarization to excite QD emission above the scattered light level. In principle, though, with proper design of components, a higher degree of pump suppression can be achieved. While it is unclear what factors contribute most to scatter from the free-space pump into the ULLW, it is likely that fabrication imperfections are to blame, which brings an undesirable degree of uncertainty to the problem. As an alternative, waveguide-based resonant pumping may provide more controllable means of minimizing waveguided pump scatter[50].

The broad linewidths observed even upon resonant excitation, due to large spectral diffusion and dephasing, limited our ability to coherently control the quantum dot and demonstrate indistinguishable single-photons. In particular, the need to co-pump the quantum dot non-resonantly with above-band light most likely contributed to an increase of the inhomogeneous linewidth particularly at higher resonant excitation[55]. It is unclear whether any of the fabrication steps were ultimately responsible for the large spectral diffusion in our devices, since the quantum dots were not characterized pre-fabrication. Screening the QD population prior to fabrication may allow identification of QDs with narrower linewidths. Deterministic positioning of single QDs within nanofabricated geometries, at sufficient distances from etched sidewalls, has been shown to be at least beneficial in preserving emission properties[32,61]. As a potential solution for improving single-photon indistinguishability, Supplementary Note 8 discusses a promising GaAs cavity, optimized with electromagnetic inverse-design and compatible with our platform, that offers, besides high coupling efficiency, a Purcell factor of ≈10, and etched sidewalls distant from the QD by more than 300 nm.

Regarding our passive photonic circuits, lower propagation losses may be achieved by employing blanket nitride growth, etch, and annealing techniques[64], as well as transverse magnetic (TM) field designs[1]. At the same time, we anticipate that a variety of on-chip passive components already demonstrated in this platform, including spiral delay lines[65], filters[66], and couplers and switches[67], can be further optimized for lower insertion losses.

Implementing all of the measures above—improving the QD-to-waveguide coupling efficiency and enhancing single-photon indistinguishability via nanophotonic design and deterministic QD positioning, and further minimizing propagation and insertion losses in passive on-chip components—will bring us closer to fully chip-integrated systems implementing practical Boson sampling and related photonic quantum information tasks with quantum advantage. We note further that the ultra-low propagation losses demonstrated here may already allow the implementation of on-chip delays for time-demultiplexing of a single quantum emitter single-photon source, to produce spatially multiplexed photons for Boson sampling similar to that demonstrated with free-space optical delays[12].

In conclusion, our results indicate high prospects for the utilization of quantum emitters as on-demand sources of single-photon in ultra-low loss, ≤1 dB/m, photonic integrated circuits, which may prove essential for the creation of scaled photonic quantum information systems on-chip.

## Methods

### Uncertainty reporting

Wherever unspecified in the text, reported uncertainties are 95% confidence intervals, corresponding to two standard deviations, resulting primarily from Type A evaluations of least-squares fits of models to data. We report other details of uncertainty evaluation as relevant.

## Estimation of misalignment between GaAs and Si₃N₄ waveguides

To estimate the misalignment between the $Si_3N_4$ and GaAs waveguides in the SEM of Fig. 1d, we calibrate the image pixel size using reference positions produced by electron-beam-lithography on the GaAs device. We then measure pixel distances between $Si_3N_4$ waveguide and GaAs support frame at various locations to determine physical distances and tilt angles. Although the uncertainty is expected to be negligible, because we do not evaluate the uncertainties related to edge thresholds, we provide conservative estimates of < 340 nm and < 0.9° for the lateral displacement and tilt angle, respectively.

## Device fabrication

Device integration involves fabricating III–V semiconductor single photon emitters in a tab-released membrane structure and employing a pick-and-place technique[24,43] to place the emitter in pockets etched in the $Si_3N_4$ waveguide upper oxide cladding. Alignment is achieved in the x-y plane using etched mechanical features in the semiconductor and waveguide upper cladding oxide pocket. Fabrication of the $Si_3N_4$ chip and the GaAs/QD devices was done in two separate runs. For the passive, ULL circuit, low pressure chemical vapor deposition (LPCVD) $Si_3N_4$ was deposited on a 100 mm silicon wafer with a 15 μm, thermally grown $SiO_2$ layer. Waveguides were patterned with a deep-ultraviolet (DUV) stepper and dry etched using an inductively coupled plasma (ICP) reactive-ion etcher (RIE) with $CHF_3/CF_4/O_2$ chemistry. A ≈ 1 μm layer of $SiO_2$ was deposited by plasma enhanced chemical vapor deposition (PECVD) using liquid tetraethoxysilane (TEOS) as a precursor of Si, followed by a high temperature anneal and chemical mechanical polishing (CMP) for planarization. Optical lithography was then used to define placement pits for the GaAs devices, aligned to buried $Si_3N_4$ waveguides. The placement pits were etched ≈ 500 nm deep into the top $SiO_2$ cladding. To better accommodate the QD devices, the pits were further trimmed with an additional optical lithography step followed by a buffered oxide etch (BOE). The visible fringes along the buried waveguide in Fig. 1a show evidence of non-uniform $SiO_2$ removal from above the $Si_3N_4$, and, potentially, also etching of the $Si_3N_4$. GaAs devices were fabricated from an epitaxially grown stack consisting of a 190 nm thick GaAs layer containing InAs QDs at the center, on top of a 1 μm $Al_{0.7}Ga_{0.3}As$ sacrificial layer. Prior to fabrication wide-field illumination photoluminescence imaging confirmed the presence of high density quantum dots emitting in the 900 nm band, with individual quantum dots addressable through a combination of spatial and spectral filtering during subsequent device characterization. Electron-beam lithography followed by $Cl_2/Ar$ ICP etching was used to define the devices on the epi-wafer, and hydrofluoric acid was used to remove the sacrificial layer. This process resulted in free-standing GaAs devices that could be picked up with a tungsten probe and placed onto the etched pits on the ULLW chip[43]. The GaAs devices and placement pits had triangular locking geometries (indicated in Fig. 1a) that enable sub-micron alignment to be achieved. The successful integration of the GaAs devices was confirmed using optical microscope as well as scanning electron microscope prior to deposition of the top $SiO_2$ cladding (see Supplementary Note 6 for details on estimating the device alignment). After device placement into the etched pits, PECVD was used to deposit a 1 μm $SiO_2$ film over the entire chip. This step created a $SiO_2$ upper cladding for the GaAs devices. Before testing, diced chip facets were polished such that the waveguide ends of the spirals were accessible via end-fire coupling.

## Cryogenic photoluminescence measurements

The fabricated devices were measured in a closed-cycle Helium cryostat at temperatures < 10 K. The sample was imaged from the top, with a micro-photoluminescence (μPL) setup implemented just above an optical window at the cryostat chamber top. Optical excitation of the QDs in the GaAs devices was also done from the top, with laser light focused to a spot of ≈ 1 μm diameter. Quantum dot emission coupled to the ULLWs was collected using a lensed optical fiber mounted on a nanopositioning stage that could be aligned to WG facets at the polished edge of the hybrid chip. The results shown here were obtained from devices that included 50:50 MMI splitters, as shown in Fig. 1. Supplementary Fig. 5 shows μPL spectra produced by one of the fabricated devices under 845 nm continuous wave (CW) laser pumping, collected separately from the two MMI output ports.

## Triggered single-photon emission measurements

We measured the lifetime of the $X_0$ line upon excitation with a < 100 fs, 80 MHz pulsed laser at 887 nm. The emission was filtered using a ≈ 500 pm bandwidth fiber coupled grating filter having efficiency of ≈ 50% and the photon counts were detected with a superconducting nanowire single photon detector (SNSPD).

To determine the purity of single photon emission, the intensity autocorrelation for the exciton line was measured using two SNSPDs in a Hanbury-Brown and Twiss configuration. Figure 3d shows the normalized photon detection coincidences, measured with a 128 ps bin size, for the $X_0$ line pumped at saturation (red dot in Fig. 3b, top). The data was fitted with a two-sided exponential decay and a $g^{(2)}(0)$ value of $0.07 \pm 0.02$ and decay parameter of $(0.85 \pm 0.02)$ ns was obtained, close to the radiative rate. This shows triggered high-purity single photon emission from the QD collected in the ULLW.

## Resonance fluorescence measurement

To observe resonance fluorescence from our device, free-space excitation was used once again, with a laser beam tuned to the $X_0$ transition in Fig. 3a. Polarization control of the excitation beam allowed us to suppress scattered pump light into the $Si_3N_4$ waveguide by as much as ≈ 25 dB while monitoring the signal on a grating spectrometer. In order for the resonance fluorescence to be observable however, it was necessary to co-excite the QD with a weak non-resonant laser at ≈ 841 nm[53]. While the non-resonant laser alone was sufficiently weak to produce negligible photon emission counts for all resonant laser powers, it enhanced the resonance fluorescence light by as much as ≈ 10 times.

The Mollow triplet spectra shown in Fig. 4c were obtained by filtering QD emission collected from the ULLW with a scanning Fabry-Perot interferometer (SFPI) with free-spectral range of 40 GHz and finesse of ≈ 200. At different resonant excitation powers, the intensity of the non-resonant co-pump was optimized to increase the resonant emission count. A ≈ 200 GHz bandwidth fiber-coupled grating filter preceding the SFPI eliminated non-resonant laser light while allowing the complete resonance fluorescence spectrum to be measured. The Mollow triplet spectra were fit, through a nonlinear least-squares method, with a function that included three Lorentzians peaks, corresponding to the center and two side-peaks of the incoherent Mollow triplet spectrum- and an additional, sharp central Lorentzian to account for the coherent resonance fluorescence signal and pump scatter. The spectral locations of the side-peaks (with 95% fit confidence intervals) are plotted as a function of pump power in Fig. 4c.

A physical model of the Mollow triplet that included effects of laser detuning and QD spectral diffusion was also used to fit the data, yielding the $T_2 < 100$ ps estimate given in the main text. A description of the model, and plots of the fits and extracted parameters are shown in Supplementary Note 9.

## Fourier-transform spectroscopy

For Fourier-transform spectroscopy, QD emission resonant with the pump laser was passed through a Mach-Zehnder interferometer (MZI)

with variable delay, then detected with an SNSPD. The MZI delay was scanned to yield an interferogram that corresponded to the first-order correlation function of the QD emission, from where the QD coherence time $T_2$ can be extracted[59]. In our experiment, the MZI was tuned to a discrete number of delay values between −0.1 ns and 0.3 ns. At each point, the MZI delay stage was dithered 5 times with an amplitude of 2 µm, giving sufficient time for the system to stabilize. Interference fringes from the latest dither were recorded and the visibility $V = (I_{max} - I_{min})/(I_{max} + I_{min})$, where $I_{max,min}$ are the maximum and minimum fringe intensities, was calculated at each point.

## Data availability
The data that support the plots within this paper and other findings of this study are available from the corresponding author upon request.

## Code availability
Code used to generate electromagnetic inverse design simulation results within this paper are available from the corresponding author upon request.

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

## Acknowledgements

We thank Edward Flagg from West Virginia University for helpful discussions regarding the resonance fluorescence results. A.C. acknowledges support under the Cooperative Research Agreement between the University of Maryland and NIST-PML, Award no. 70NANB10H193. J.D.S. acknowledges the program of quantum sensor core technology through IITP (MSIT Grant No. 20190004340011001). D.E. and H.L. acknowledge support from the NSF (1936314QII-TAQS) and AFOSR program FA9550-16-1-0391. H. L. acknowledges the support of the Natural Sciences and Engineering Research Council of Canada (NSERC), the National Science Foundation (NSF, Award no. ECCS-1933556), and of the QISE-NET program of the NSF (NSF award DMR-1747426). E.G.M. acknowledges support from the São Paulo Research Foundation (FAPESP) Grant 2021/10249-2.

## Author contributions

M.D. and E.G.M. performed electromagnetic design of the hybrid devices. J.C., D.E., R.M., D.J.B., K.S., and M.D. conceptualized the hybrid chip pick-and-place assembly method. J.S. provided the molecular beam epitaxy-grown quantum dot sample. R.M. designed and fabricated the $Si_3N_4$ photonic circuit chip. H.L. devised and performed post-fabrication adjustments to the $Si_3N_4$ on-chip devices. B.G. fabricated the GaAs devices. J.C. and H.L. performed pick-and-place assembly of the hybrid device. V.A. provided superconducting nanowire single-photon detectors. A.C. performed the waveguide propagation loss and single quantum dot characterization measurements, analyzed the data and produced figures for the manuscript. A.C. and M.D. wrote the manuscript, with input from all authors. M.D., K.S., D.J.B., D.E., and J.C. supervised the project. All the authors contributed and discussed the content of this manuscript.

## Competing interests

The authors declare no competing interests.
