## [Peer Review File · Nature Communications]

Ultra-low loss quantum photonic circuits integrated with single quantum emittersREVIEWER COMMENTS

Reviewer #1 (Remarks to the Author):

Chanana et al. report the ultra-low loss integration of quantum dot single photon sources in silicon nitride waveguides.

Technologically, the findings are rather intriguing, since the authors demonstrate that waveguide losses are more than two orders of magnitude less than those reported for systems that combine single photon emitters with waveguide.

The authors developed a comparable integration system a few years ago; the primary contributions of the present work are decreased waveguide losses and resonant excitation, the latter of which was achieved on chip with superconducting single photon detectors.

- Mario Schwartz, Ekkehart Schmidt, Ulrich Rengstl, Florian Hornung, Stefan Hepp, Simone L. Portalupi, Konstantin Ilin, Michael Jetter, Michael Siegel, and Peter Michler. "Fully On-Chip Single-Photon Hanbury-Brown and Twiss Experiment on a Monolithic Semiconductor–Superconductor Platform."

Despite the fact that the waveguide loss is quite small in comparison to prior work, the integrated single photons source has a limited net increase in brightness. Even in the simulated configuration, the high aspect ratio of the waveguide, which may operate as a polarizing filter for TM modes, further reduced the coupling efficiency between the emitter and the waveguide to 31%.

I find the results really intriguing, particularly the resonance excitation. I have suggestions for the authors to enhance the impact of the work and its novelty in order to warrant publishing in nature communication.

- Compare the indistinguishability of the emitted photons from various excitation schemes, i.e., above band vs resonant excitation, this is extremely desired for waveguide-integrated sources.
- How might the spectral diffusion described in the paper influence the indistinguishability of future QD pulses? Experimental measurement of HOM interference should be provided.
- The authors should provide a strategy for implementing this resonant excitation technique with an on-chip supplied pump. As one of the promises of hybrid photonic integration is the realization of large-scale systems, this is incompatible with free space pumping. A graphic of simulations alone would suffice.
- The authors should provide a method for reducing the flickering of the quantum dot emission. This may be associated with surface charges in the etched waveguide and photonic crystal reflector. Is this always a technological limitation?

Reviewer #2 (Remarks to the Author):

The work from Ashish Chanana et al. reports on efficient integration of III/V quantum light sources on ultra-low loss SiN waveguides. Ultra-low loss photonic integrated circuits are extremely important to bring scalability to quantum photonics applications since photon losses is one of the main bottlenecks.

Low loss circuitry is not totally new but have never been combined with high quality quantum dots. However, the authors do not use the ultra-low waveguides to perform an experiment which would have not been possible with higher losses. Such a smoking gun experiment would clearly help the manuscript to reach even broader audience. Also the authors did not show how their approach is scalable, the main reason for integration. Showing integration of several emitters and interference on a single chip would have been a real breakthrough but that would require a tuning technique which was not realized. Nonetheless the authors achieved an important technological step, crucial for the current photonic quantum technology developments. Therefore, I believe the current manuscript warrants publication in Nature Communications despite the mentioned short comings. I just hope that the same authors do not send a multi-emitter interference experiment out for review in the next coming months since I would rather have liked a combined manuscript. I truly believe that their integration approach has great potential and can be a game changer to bring quantum light sources back on the radar of photonic quantum computing.

I would appreciate it if the authors address the following points prior publication in Nature Communication:

- How do you envision reconfigurable linear optical networks in SiN at cryogenic temperatures? Apart from MEMS I am not familiar with techniques compatible with low temperatures.

⁷⁵ the appearance of the Mollow triplet in the QD emission spectrum. Such a feature is a signature
⁷⁶ of dressed states emerging from the coupling of a two-level system to a strong coherent excitation
⁷⁷ field^{25,26}, and is of both scientific and technological importance^{27,28}.

- I would like to see an explanation why the Mollow triplet has technological importance. Reference 27 states that the cascade is not deterministic, there can always be Rayleigh emissions in between the cascade. For most photonic quantum technologies, I do not see the need of Mollow triplet generation and rather on-demand generation of indistinguishable photons, which could have been measured using pulsed RF. Can the authors comment why no pulsed RF and no HOM experiment was included?

- Fig2a. Why do you have this arrow on the pump? To indicate that you can scan the pump? Maybe then add as second perpendicular arrow to show scanning in x and y?

- Fig3d. "Back-scattered light intensity as a function of propagation length and RoC along the 1 m, 2 m, and 3 m spirals". Not 100% clear if 1m or 3m is the top image. I would add the information in the graphs respectively.

- What is the coupling efficiency from the waveguide mode to the fiber? Was a specific waveguide tapering used to increase this value?

- Can you clarify in the main text if you have 4% coupling efficiency to the detectors or system detection efficiency? This only becomes clear from the supplemental.

- Fig 4a. It is unclear what electron configurations are labeled X1, X2, X3. Typically, a QD has a trion state a neutral exciton and possible an XX state. Can the authors label the lines based on power, polarization, correlation measurements? Otherwise one could imagine that these 3 lines stem from different quantum dots.

- Fig4c. Where does the large noise stem from in the TCSPC measurement? The SSPDs should yield a much lower noise level. Why is the decay double exponential?

- “At excitation powers significantly below the saturation level, elastic resonant Rayleigh 195 scattering dominates, and a spectrally narrow emission line is observed⁴⁴.” (Matthiesen, C., Vamivakas, A. N. & Atatüre, M.). It is not an emission line, it is coherent scattering which does not anti-bunch (see Hanschke et al. Phys. Rev. Lett. 125, 170402 (2020), which might be the better reference, so the misconception of sub-natural linewidth single photons is not further spread).

233 for details). A reference visibility trace, obtained for non-resonant pumping, is shown in panel
 234 **i** of Fig. 5d. The trace is fitted with a Gaussian, which indicates spectral diffusion, and yields
 235 $T_2 = (0.053 \pm 0.003)$ ns. Panels **ii** to **iv** in Fig. 5d are visibility traces for resonance fluorescence
 236 for varying excitation powers, as indicated by the Rabi frequencies. It is worth noting that at the
 237 higher powers Rabi oscillations are visible, which are reasonably well reproduced by the model⁵⁵.
 238 The corresponding coherence times are **i** $T_2 = (0.10 \pm 0.1)$ ns, **ii** $T_2 = (0.07 \pm 0.01)$ ns and **iii**
 239 $T_2 = (0.09 \pm 0.01)$ ns, longer than for the non-resonant excitation values. The coherence dynam-

- The labeling with “i” is somewhat confusing. If 5d i is non-resonant why is i T_2 larger than ii and iii. But you write that i, ii, iii are longer than for the non-resonant values, which should be depicted in 5d i. Maybe it is only me not getting it, but probably you can reduce potential of getting confused.

- “by > 3 dB/m, than the lowest reported for any foundry-compatible quantum PIC to date.” Please check works from QuiX and Xanadu both using SiN chips with ULLW. I am unsure if these companies have lower values but it is worth checking.

- The fabrication and alignment of such an adiabatically tapered structure is very impressive. (Already published by the same authors) It would be great if either a

simulation of the impact of misalignment on the coupling efficiency is given in the supplemental or if a reference to a previous publication with the values is included.

- References: Somaschi, N. et al. Near-optimal single-photon sources in the solid state. There is another paper published slightly earlier: Phys. Rev. Lett. 116, 020401

- Supplemental TABLE S1, missed values from Hyperlight (Loncar) and as mentioned before recent QuiX and Xanadu values. Maybe their current values are published somewhere?

SiN/InP (nanowire integration) is also missing in the table, probably also with very high losses as far as I can tell.

- Supplemental RF fitting: Did you consider the coherently scattered laser on top of the Rayleigh peak in the Mollow triplet fitting?

Reviewer #3 (Remarks to the Author):

The authors report on the integration of a GaAs nanophotonic device, which embeds InAs quantum dots, on top of a photonic circuit made of Si₃N₄. Together with the characterization of the Si₃N₄ chip, proving low propagation losses, the authors perform quantum dot measurements to prove the injection of single photons in the silicon based circuit. Furthermore, they excite the quantum dot with a resonant laser in the strong driving regime, observing the Mollow triplet spectrum.

The novelty point of this work is based on the low achieved losses in the Si₃N₄ waveguides, while for the integration of the nanophotonic device they employed pick-and-place techniques already proven valid for quantum dots as well as other material platforms.

Despite the waveguide technology is impressive, I do not think that this work presents enough novelty and impact to be published in Nature Communications. I would rather support the publication in a more specialized journal.

In more details: as said in the abstract, one important goal would be the combination of bright sources with ultra-low loss waveguides. While low loss has been achieved, this value does not play a role in the experiments, since the nanophotonic device is coupled to a straight waveguide (which includes a beamsplitter that is not used in the experiment) instead to a complex photonic chip as shown in Fig. 1 (even the spirals are not used in combination with the QD). In addition, the QD-to-ULLW coupling is rather modest (below 7%) being limited by design and transfer imperfections. Chanana et al. provide a perspective on how to improve the performances but these approaches may improve some performances at the expense of others, as discussed below.

I here provide additional comments which can help strengthening the message for a specialized audience.

- 1) Figure 1 looks very nice but it is not needed in the paper: it provides a perspective but it does not add useful information for the reader. In particular because, as I understand the paper, the quantum dot is coupled to a straight waveguide, while the spirals are made on another chip and are characterized with a laser (figure 3).
- 2) The coupling efficiency from the GaAs to the Si₃N₄ is rather modest at this stage (between 4 and 7 %). Apart from the dipole moment orientation, the limited coupling is attributed to misalignment and geometrical imperfections. Therefore, while the Si₃N₄ photonics is of very high quality, the performances are limited by the rest of the processing, so limiting the overall impact.
- 3) Following the previous point, if a different dipole orientation would increase the coupling up to 30% (as simulated in the SI), the question arises on the fabrication reproducibility. In other words, how many samples have been fabricated? Statistically, some of them would have the correct dipole orientation increasing the performance.
- 4) In paragraph II, it is said that a mode transformer needs a small spacing (between transformer and waveguide), which has not been used in the current geometry. Can the achieved low propagation losses be maintained even modifying the WG geometry to achieve an increased in-coupling? Since it is said that it can be done to reach 93%, the question may arise why it has not been done, getting an efficiency of 7%.
- 5) In paragraph III, linear losses for the spirals were found. Why losses for the 3 meter spiral are almost three times smaller than for the 2 meter one (in terms of dB per meter)? Is somehow the fabrication more effective for long spirals or is it related to low sample statistics?
- 6) In paragraph IV, the used quantum dot is described but some important information is missing: which emitter transitions are X1, X2 and X3? In particular X1, is it a neutral exciton, a charged exciton or a biexciton?
- 7) Again in paragraph IV, the quantum dot photons are sent through a MMI beamsplitter but it seems that for the Hanbury-Brown and Twiss setup an external beamsplitter is employed. This would mean that the MMI is only an additional source of losses (around 45% as in the SI). As said before, the experiment does not really make use of the low loss Si₃N₄ for performing the measurements.
- 8) The authors refer in the text to Figure 1 a or b when they mean figure 2. In the methods section,

they refer to figure 4e (it should be d) and fig. 4b (should be a).

9) In figure 4 (and describing text) a $g(2)$ of 7% is reported. Since the filter used has 500 pm bandwidth (as written in the SI), the $g(2)$ is still good, despite the underlying background of X1 within 500pm range. Has the background being subtracted? It can be helpful to mention it.

10) The T1 value reported for resonant excitation has been obtained by fitting the $g(2)$ in figure 5. It would be useful to report which excitation power has been used and if deconvolution has been performed in the fit in fig.5b.

11) Regarding figure 5d, the authors should comment on why using a Gaussian fit instead of a Voigt (which would include the Lorentzian component from the lifetime).

12) In paragraph VI, approaches to achieve better performances are discussed, but it is not clear if these improvements will come at the expense of other parameters. For example, increasing the in-coupling may need different waveguide geometries. How the losses will be impacted?

13) The discussion in paragraph VI on polarization entangled photons from a quantum dot is a bit misleading. To my understanding, the used waveguides support only one polarization, which would impact the polarization entanglement. Using waveguides which support both polarizations may impact the losses (which are the main novelty point of the work).

In conclusion, I appreciate the technological steps forward made by the authors but this study should be published in a more technical journal. The proposed changes may provide additional interesting information for a specialized audience.

REVIEWER COMMENTS

We thank the reviewers for their detailed and constructive feedback, which has helped us to greatly improve our manuscript. Below we provide a point-by-point response to all referee comments, with the referee remarks in black text and our responses in blue text. At the end, we include a list of changes made to the manuscript.

Reviewer #1 (Remarks to the Author):

Chanana et al. report the ultra-low loss integration of quantum dot single photon sources in silicon nitride waveguides.

Technologically, the findings are rather intriguing, since the authors demonstrate that waveguide losses are more than two orders of magnitude less than those reported for systems that combine single photon emitters with waveguide.

The authors developed a comparable integration system a few years ago; the primary contributions of the present work are decreased waveguide losses and resonant excitation, the latter of which was achieved on chip with superconducting single photon detectors.

- Mario Schwartz, Ekkehart Schmidt, Ulrich Rengstl, Florian Hornung, Stefan Hepp, Simone L. Portalupi, Konstantin Ilin, Michael Jetter, Michael Siegel, and Peter Michler. "Fully On-Chip Single-Photon Hanbury-Brown and Twiss Experiment on a Monolithic Semiconductor–Superconductor Platform."

We thank the reviewer for pointing out this important paper, which we unfortunately failed to properly reference in our manuscript. We note that, while resonance excitation was indeed demonstrated in this paper, observation of the Mollow triplet as in our manuscript was not reported. The works also differ in terms of the platform and waveguide loss, as the referee has noted. We have implemented the following comment to the end of the first paragraph of the Resonance Fluorescence section of the manuscript:

"We note that resonance fluorescence has also been observed without polarization filtering in AlN circuits with integrated Ge-vacancy quantum emitters in diamond [Wan, Nature 2020], and control of pump polarization alone was sufficient to allow observation of waveguide-coupled resonance fluorescence with on-chip SNSPDs in ref. [Schwartz, Nano Letters 2018]."

Despite the fact that the waveguide loss is quite small in comparison to prior work, the integrated single photons source has a limited net increase in brightness. Even in the simulated configuration, the high aspect ratio of the waveguide, which may operate as a polarizing filter for TM modes, further reduced the coupling efficiency between the emitter and the waveguide to 31%.

We thank the reviewer for the careful read of our manuscript. We emphasize that, while the devices experimentally demonstrated in the present manuscript have indeed featured low (< 7

%) coupling efficiency into the ultra-low loss waveguide, the main reason for such low efficiency is the misalignment of the quantum dot's electric dipole transition with respect to the TE GaAs waveguide mode, as is evident in Table S2 of the supplemental information.

We point out, though, that this is not an issue that is inherent or unique to our platform, but rather one that is ubiquitous in photonic devices with functionality enabled by single quantum emitters, if the latter are not deterministically grown or placed within the device. Importantly, in the present work (and in contrast with our work published in Nano Letters 39 7164 (2019)), we have not attempted to create devices with deterministically positioned (and oriented) quantum dots. We therefore argue that the low efficiency observed here is not fundamental, and is absolutely not illustrative of the full potential of our platform.

A different issue, on the other hand, regards the maximum theoretical 31 % extraction efficiency that would be achievable with the fabricated devices, provided the QD can be properly aligned and oriented within the geometry. Indeed, the specific GaAs geometry that was experimentally tested had not been designed for high coupling efficiency, and had a number of features that altogether contributed to such overall low values.

More specifically, as illustrated in the schematic of Fig. R1_1 below, the geometry launches emitted photons with an efficiency β into a forward traveling straight waveguide mode that is spatially confined in GaAs ('GaAs mode' in the figure). A mode transformer then converts such GaAs waveguide mode into a Si3N4 mode with efficiency η_{MT} , resulting in a total overall QD-to-Si3N4 waveguide $\eta_{QD_ULLW} = \beta * \eta_{MT}$. Finite Difference Time-domain (FDTD) simulations of the tested device geometry yielded $\beta = 0.88$, $\eta_{MT} = 0.35$, indicating a major detrimental contribution from the mode transformer.

As shown in Section VII of the revised SI, however, a more carefully designed mode transformer may be implemented to yield $\eta_{MT} = 0.92$, which would readily provide an overall efficiency $\eta > 0.8$. Further improvement could be achieved in a variety of ways, for instance by increasing β . We refer the referee to our recent work placed in the arXiv (Melo et al., arXiv:2206.01043), which describes geometries providing $\beta > 0.92$ with Purcell factors of > 20 , in geometries in which the quantum dot is located > 300 nm away from any etched sidewalls. Such a design is completely compatible with the present platform, and has the potential to circumvent, or at least improve, via both the Purcell enhancement and distancing from etched sidewalls, parameters such as single-photon indistinguishability, spectral diffusion and blinking. To illustrate the compatibility of such geometries with our platform, Fig R1_2 below shows a GaAs geometry embedded in SiO2 that was designed with the same inverse-design approach as in the above arXiv manuscript. The simulated geometry yields $\beta=0.91$ into the GaAs output waveguide mode, with a Purcell factor of 10. This result has been included in Section VIII in the revised SI.

Fig. R1_1: Schematic of the hybrid single quantum dot single-photon source. Two cross sections are shown, where the relevant features are highlighted. Cross-sectional views of the modes supported in two different sections of the device along z are shown at the bottom. The quantum dot emits into the GaAs mode with efficiency β , which is transformed into the Si₃N₄ mode by the mode transformer, with efficiency η_{MT} . The quantum dot coupling efficiency into the Si₃N₄ waveguide mode is $\eta_{QD-ULLW} = \beta \eta_{MT}$.

Fig. R1_2: Example GaAs nanophotonic geometry, embedded in SiO₂, that provides $b=0.91$ into the output GaAs waveguide mode, same as in Fig. R1, with a Purcell radiative enhancement factor of 10. As in Fig. R1, light from such GaAs mode can be converted into the Si₃N₄ mode with the optimized mode transformer discussed in the SI, yielding an overall efficiency $\eta \sim 0.85$. The top figure shows the refractive index distribution of the device, whereas the bottom shows the electric field distribution for an optimally positioned quantum dot.

Further improvements could potentially be achieved by employing a high quality factor cavity evanescently coupled directly to the Si₃N₄ waveguide. As shown in Katsumi et al. *Optica* 6 786 (2019), $\eta > 99\%$ can potentially be achieved, albeit over a relatively narrow wavelength band.

We note lastly that the issue of supporting TE modes and operating as a TM mode filter is discussed in Bauters, J. F., et al. (2013). "Ultralow-Loss Planar Si₃N₄ Waveguide Polarizers." *IEEE Photonics Journal* 5(1): 6600207-6600207. Distinguishing between modes by introducing higher loss for one mode than the other can be achieved by adjusting the bending radius such that TM bend losses are much higher than TE. Therefore it depends on the type of structure that one is coupling to in order for there to be a mode filter. This is not a feature of the high aspect ratio waveguide by itself. This is an important issue to study in future work and is outside the scope of the current manuscript.

I find the results really intriguing, particularly the resonance excitation.

We thank the reviewer for the positive comment.

I have suggestions for the authors to enhance the impact of the work and its novelty in order to warrant publishing in nature communication.

1 - Compare the indistinguishability of the emitted photons from various excitation schemes, i.e., above band vs resonant excitation, this is extremely desired for waveguide-integrated sources.

The low coherence time of our quantum dot source, even under resonant excitation ($T_2 < 100$ ps $< T_1 \sim 630$ ps), makes such a comparison difficult, as the visibility of interference given by $T_2/(2T_1) < 0.1$ is low in all cases. In principle, 'post-selected' indistinguishability measurements, i.e., by time-resolving the Hong-Ou-Mandel interference dip, could be used to differentiate the cases, but the coherence time T_2 is comparable to the ~ 100 ps timing resolution of our setup, so this approach is also limited.

However, we agree that a discussion of the performance of the system under different pumping conditions is warranted. First, we note that panel i in Fig. 5d (Fig. 5e in the revised manuscript) already presents a visibility trace for quasi-resonant p-shell pumping, which leads to an estimated $T_2 \sim 50$ ps. This value is approximately half of that observed with resonant excitation.

Adding to this result, in Section XII of the revised Supplementary Information we show a comparison of high-resolution emission spectra, obtained with a scanning Fabry-Perot interferometer, for the QD under non-resonant (wetting layer), quasi-resonant (p-shell) and resonant excitation. Under non-resonant and quasi-resonant excitation, significant spectral broadening was observed, captured primarily by the Gaussian linewidth obtained in a Voigt fit. In comparison, the resonant emission spectrum exhibits a characteristic Mollow triplet shape, which extends over a comparable spectral extent as the non-resonant spectra. The central peak is furthermore significantly narrower, and is well fitted by a Lorentzian lineshape, as shown in Fig S10 .

We note also that the variation of spectral diffusion under different excitation types is also captured by the Mach-Zehnder visibility traces shown Fig 5d and 5e. Here for p-shell pumping (Fig. 5e), the visibility trace is fit to the sum of a Gaussian and two-sided exponential, where the ratio of Gaussian and exponential components are comparable in amplitude. In comparison, for resonant excitation at low excitation powers, the visibility traces fit well to a two-sided exponential. At high resonant laser powers however, the fit is predominantly Gaussian. The data has been provided in Table S3 of Supplementary Information section X. This effect also indicated spectral diffusion under quasi-resonant excitation and at high powers for resonant excitation.

2 - How might the spectral diffusion described in the paper influence the indistinguishability of future QD pulses? Experimental measurement of HOM interference should be provided.

As noted above, the coherence time of our QD source, even under resonant excitation, is rather low and likely a consequence of spectral diffusion. Having measured T_2 and T_1 for the system, we can make an estimate of the best pulsed HOM indistinguishability possible at this point as $V = T_2/(2 \cdot T_1) \sim 0.1$. Going forward, a combination of approaches, fully compatible with our

process, should be possible to address this situation. This could include the use of Al₂O₃ atomic layer deposition to passivate the etched sidewalls (see e.g., Liu et al., Phys. Rev. Appl., 9 064019 (2018)), the development of designs that keep etched surfaces sufficiently far away from the QD (e.g., Melo et al., arXiv:2206.01043) provided that QD location techniques are used in the fabrication process (Schnauber et al., Nano Letters 39 7164 (2019)), and the incorporation of electrical gating techniques to control the charge environment surrounding the QD (e.g., Kirsanske et al., PRB 96 165306 (2017)) .

3 - The authors should provide a strategy for implementing this resonant excitation technique with an on-chip supplied pump. As one of the promises of hybrid photonic integration is the realization of large-scale systems, this is incompatible with free space pumping. A graphic of simulations alone would suffice.

We thank the reviewer for the important comment. Indeed, we believe waveguided resonant pumping is likely the best way forward to scaling the platform. We can envision various ways in which this could be accomplished. One particularly interesting way is that of Uppu et al., Nat. Comms 11 5198 (2020), which employs a first-order GaAs waveguide mode to pump the quantum dot, followed by a mode filter and bent waveguide that greatly extinguish the pump. While in that paper the source is entirely implemented in a GaAs circuit, a hybrid version may be implemented in our platform, replacing the grating couplers with waveguide mode transformers. A schematic is provided in Fig. R1 below. While a better understanding of the pumping scheme can be gained from the paper directly, here we provide a brief explanation for the referee's benefit.

The resonant laser is launched from a Si₃N₄ waveguide into a single-mode GaAs waveguide (SMW1) with high efficiency, via an adiabatic mode transformer (MT1) such as discussed in our response above to the referee's second comment. A fast GaAs waveguide width taper (T1) excites the fundamental and first order mode of the multi-mode GaAs waveguide (MMW). The photonic crystal reflector (PC) blocks the fundamental mode, but lets the first-order mode pass through and excite the quantum dot. The quantum dot emits with high efficiency into the fundamental mode, and with considerably lower efficiency into a high-order mode. An adiabatic taper (T2) transitions the multimode waveguide into a single-mode guide (SMW2). Since the pump is initially carried by a first-order mode of MMW, it leaks out of the SMW2, which only supports a fundamental mode. To further eliminate residual pump power, a curved section of the SMW2 waveguide (PF) is employed, which significantly enhances pump leakage, thereby acting as a pump filter. The emitted quantum dot light, on the other hand, is still carried by the fundamental mode of SMW2. Finally, an adiabatic mode transformer (MT2, same as MT1) is used to transfer the quantum dot emission, carried by the fundamental SMW2 mode, into the Si₃N₄ collection waveguide.

Fig. R1_3: Schematic of waveguide-based resonant scheme, based on Uppu et al., Nat. Comms 11 5198 (2020), which may be implemented in our platform. While Uppu et al. employ grating couplers to launch the pump into the GaAs waveguides, and to extract the quantum dot resonance fluorescence from their hosting waveguides, in our scheme the grating couplers are replaced by mode transformer to / from the Si₃N₄ waveguides. All other functional elements on this scheme, associated with excitation, pump suppression, and emission collection (photonic crystal reflector, multimode - to single-mode transitions, pump laser filter) can be implemented in our platform with trivial modifications.

An improved scheme utilizing a photonic crystal waveguide, recently reported in by Xiaoyan Zhou et al 2022 *Quantum Sci. Technol.* 7 025023 (2022) could be implemented in the same fashion. At the same time, we point out that various alternative geometries could also be utilized, and that is a current subject of our investigation.

4 - The authors should provide a method for reducing the flickering of the quantum dot emission. This may be associated with surface charges in the etched waveguide and photonic crystal reflector. Is this always a technological limitation?

We thank the reviewer for the important question. Although the origin of the flickering is not entirely clear, it most likely has to do with charge fluctuations, either at traps near the quantum dot, or from etched surfaces. Notably, however, the GaAs waveguide had a width of 300 nm, consistent with our observation of significant linewidth broadening and blinking, in a pre/post fabrication comparison reported in PR Applied 9, 064019 (2018). A potential solution for this issue would be to create a photonic geometry in which the quantum dot is distant by > 300 nm from any etched sidewalls, and which efficiently outcouples quantum dot emission efficiently into

a single TE GaAs waveguide mode. One candidate is the cavity designed in our recent arXiv preprint (<https://arxiv.org/abs/2206.01043>), which provides 92 % outcoupling efficiency into the fundamental TE GaAs waveguide mode with a Purcell factor of > 20 . It is worth noting that the Purcell factor can likely be leveraged to enhance the emitter's indistinguishability.

Alternatively, Liu et al., PR Applied 9, 064019 (2018) also presented strong evidence that surface passivation by atomic layer deposition Al₂O₃ has good potential to reign in quantum dot spectral diffusion, and could be explored further, perhaps in combination with a cavity design as discussed above.

We note finally that research groups that have demonstrated highly coherent quantum dots have typically had the emitters embedded in p-i-n diode structures (e.g., Kirsanske et al., PRB 96 165306 (2017), Pedersen et al., 7 2373 (2020)). This possibility likely allows some stabilization of the quantum dot's charge environment, alongside charge and wavelength tunability. Such types of structures could potentially also be implemented in our hybrid platform, with minimal changes .

Reviewer #2 (Remarks to the Author):

see uploaded file.

Sorry for being 1 week late due to personal reasons.

all the best with this impressive platform!

We thank the reviewer for the highly positive and encouraging comment.

The work from Ashish Chanana et al. reports on efficient integration of III/V quantum light sources on ultra-low loss SiN waveguides. Ultra-low loss photonic integrated circuits are extremely important to bring scalability to quantum photonics applications since photon losses is one of the main bottlenecks. Low loss circuitry is not totally new but have never been combined with high quality quantum dots. However, the authors do not use the ultra-low waveguides to perform an experiment which would have not been possible with higher losses. Such a smoking gun experiment would clearly help the manuscript to reach even broader audience. Also the authors did not show how their approach is scalable, the main reason for integration. Showing integration of several emitters and interference on a single chip would have been a real breakthrough but that would require a tuning technique which was not realized. Nonetheless the authors achieved an important technological step, crucial for the current photonic quantum technology developments. Therefore, I believe the current manuscript warrants publication in Nature Communications despite the mentioned short comings. I just hope that the same authors do not send a multi-emitter interference experiment out for review in the next coming months since I would rather have liked a combined manuscript. I truly believe that their integration

approach has great potential and can be a game changer to bring quantum light sources back on the radar of photonic quantum computing.

We appreciate the referee's positive evaluation of our work and appreciate the perspective regarding future experiments on taking explicit advantage of the ultra-low-loss waveguides and achieving multi-emitter interference on a chip. While we agree that these are both important advances to pursue, there are non-trivial steps needed to move to such experiments. We appreciate that the referee has endorsed publication in Nature Communications despite such experimental results not being currently available.

I would appreciate it if the authors address the following points prior publication in Nature Communications:

1 - How do you envision reconfigurable linear optical networks in SiN at cryogenic temperatures? Apart from MEMS I am not familiar with techniques compatible with low temperatures.

We anticipate that, besides MEMS (e.g., as covered by Errando-Heranz, JSTQE 8200916 (2019)), stress-optical phase modulation via e.g., PZT actuators placed on top of select portions of Si₃N₄ waveguides or resonators, as recently demonstrated by our group in arXiv:2206.09245 (<https://arxiv.org/abs/2206.09245>) may be suitable. The ultra-low losses are preserved, the process is wafer-scale, and the actuation is fast (~ 20 MHz), with low power consumption, and would likely not degrade significantly at cryogenic temperatures. A potential alternative, also using the stress-optical effect, is the AlN/Si₃N₄ platform demonstrated in Nature Photonics 16 59 (2022), though the losses reported there were close to 40 dB/m. We have added this information to the main text, as we believe it further highlights the potential of our platform.

⁷⁵ the appearance of the Mollow triplet in the QD emission spectrum. Such a feature is a signature
⁷⁶ of dressed states emerging from the coupling of a two-level system to a strong coherent excitation
⁷⁷ field^{25,26}, and is of both scientific and technological importance^{27,28}.

2 - I would like to see an explanation why the Mollow triplet has technological importance. Reference 27 states that the cascade is not deterministic, there can always be Rayleigh emissions in between the cascade. For most photonic quantum technologies, I do not see the need of Mollow triplet generation and rather on-demand generation of indistinguishable photons, which could have been measured using pulsed RF.

We agree with the referee that there is no need for observation of the Mollow triplet in applications where just deterministic single-photon generation is required. As the referee pointed out, pulsed resonance fluorescence would suffice for that. We did not mean to suggest that the technological relevance of the Mollow triplet would be in terms of the same type of

photonic quantum systems such as depicted in Fig.1. Rather, as discussed in refs.[27,28], there is potential for the creation of heralded sources of single-photons or photon bundles from filtered, cascaded Mollow triplet emission. The strong correlations found between photon pairs could be explored for e.g. multiphoton spectroscopy studies [27,28], or for the performance of Bell tests, which is central to quantum communications [Peiris et al., PRL 118 030501 (2017)].

We have altered the manuscript to reflect this information:

Such a feature is a signature of dressed states emerging from the coupling of a two-level system to a strong coherent excitation field [25,26], which is not only of scientific relevance, but which also offers prospects for the development of sources of single correlated photon pairs or photon bundles, which may find applications in e.g., multiphoton spectroscopy [27,28] or quantum communications [Peiris et al., PRL 118 030501 (2017)].

3 - Can the authors comment why no pulsed RF and no HOM experiment was included?

A combination of low overall counts, limited pump suppression and short coherence times prevented us from observing measuring pulsed HOM. We believe the same factors would have rendered a CW HOM measurement challenging - in particular, though a CW HOM measurement doesn't necessarily require T2 to be large compared to T1 in order to resolve a dip that is beyond the classical level, it does require T2 to be sufficiently large in comparison to the system resolution to be able to resolve the HOM dip without unreliable deconvolution. Unfortunately, in our case, the QD T2 is close to that of our system timing resolution.

We have on the other hand provided pulsed RF measurements in the revised Supplementary Information's Section XI. In our measurements we show pulsed $g^{(2)}(0) = 0.18 \pm 0.03$, hence showing triggered single photon emission under resonant excitation.

4 - Fig2a. Why do you have this arrow on the pump? To indicate that you can scan the pump? Maybe then add as second perpendicular arrow to show scanning in x and y?

Thank you for the comment, the arrow indicates the polarization of the pump, which was used to control the latter's suppression at the waveguide. We have explicitly addressed this fact in the figure caption.

5 - Fig3d. "Back-scattered light intensity as a function of propagation length and RoC along the 1 m, 2 m, and 3 m spirals". Not 100% clear if 1m or 3m is the top image. I would add the information in the graphs respectively.

Thank you for the suggestion, we are now indicating the lengths directly on the respective figures.

6 - What is the coupling efficiency from the waveguide mode to the fiber? Was a specific waveguide tapering used to increase this value?

The coupling efficiency was estimated to be 0.63 dB/facet into a ~ 2.0 μm Gaussian spot, based on an overlap integral calculation, with no waveguide tapers. The weak modal confinement is the primary reason for such high values. Experimentally, however, cutback measurements done on straight waveguides have suggested a coupling efficiency of ~ 2 dB / facet, without any tapering. The discrepancy with simulations is likely due to modal mismatch with the tapered optical fiber that was utilized. This information is provided in Section V of the SI, which regards estimation of the coupling efficiency.

7 - Can you clarify in the main text if you have 4% coupling efficiency to the detectors or system detection efficiency? This only becomes clear from the supplemental.

Thank you for the suggestion, we have added this information to the main text. Indeed, 4 % is the lower bound for the quantum dot-to-Ullwg coupling. The detector efficiency was ~ 71 %, whereas the system efficiency is ~ 11 %, as stated in Section V of the SI.

8 - Fig 4a. It is unclear what electron configurations are labeled X1, X2, X3. Typically, a QD has a trion state a neutral exciton and possible an XX state. Can the authors label the lines based on power, polarization, correlation measurements? Otherwise one could imagine that these 3 lines stem from different quantum dots.

We thank the reviewer for this comment. We have indeed performed a series of measurements on the three lines, in an attempt to determine their respective excitonic characters. This information is now provided in Section IV of the revised SI. Cross-correlation measurements between the three lines all have featured antibunching dips, which indicates they all belong to the same dot. The observed lifetimes for C1, C2 and X0 were 0.75 ns, 0.45 ns and 1.1 ns respectively. The radiative lifetime of C2 being approximately half of the X0 line indicates that the C2 emission line could be the bi-exciton, while C1 could be another charged excitonic transition. In addition, the slope of the power series was obtained as 1.4, 1.56, and 1.06. The slope of X0 close to 1 suggests that the transition is likely the neutral exciton. However, the asymmetric bunching peak that is characteristic of cascaded emission (e.g., between XX and X) has not been observed in any cross-correlation, so we believe C1 and C2 are both charged excitonic states.

9 - Fig4c. Where does the large noise stem from in the TCSPC measurement? The SSPDs should yield a much lower noise level. Why is the decay double exponential?

We thank the reviewer for the careful read of our data. The data shown in Fig. 4 are all for non-resonant excitation, which results in an emission spectrum with a broad pedestal, as shown in Fig. 4 a, likely due to multi-excitonic transitions. To obtain the lifetime and $g(2)$ measurements shown in Figs. 4c and 4d, respectively, the X0 transition was filtered with a ~ 500 pm bandpass

filter, which allowed some of the pedestal to pass through. The imperfect filtering likely has a significant contribution to the observed noise background, as well as the secondary, slower exponential decay. A potential mechanism for the slow decay, however, may also be occasional shelving of the excited state into a slower decay state, due to a fluctuating charge environment.

10 - "At excitation powers significantly below the saturation level, elastic resonant Rayleigh 195 scattering dominates, and a spectrally narrow emission line is observed⁴⁴." (Matthiesen, C., Vamivakas, A. N. & Atatüre, M.). It is not an emission line, it is coherent scattering which does not anti-bunch (see Hanschke et al. Phys. Rev. Lett. 125, 170402 (2020), which might be the better reference, so the misconception of sub-natural linewidth single photons is not further spread).

We thank the reviewer for this important comment, which we agree with. We edited the text to read:

"At excitation powers significantly below the saturation level, elastic resonant Rayleigh scattering dominates the observed spectrum, featuring an apparent linewidth narrower than the emitter's radiative limit. While observation of antibunching of such signal has initially been reported (Mathiessen et al.), its statistics has been shown to change significantly upon narrow spectral filtering (Hanschke et al. PRL 2020, Phillips et al., PRL 2020), a behavior that has been explained as interference between coherent scattering and weak incoherent emission (Hanschke et al., PRL 2020).

²³³ for details). A reference visibility trace, obtained for non-resonant pumping, is shown in panel
²³⁴ **i** of Fig. 5d. The trace is fitted with a Gaussian, which indicates spectral diffusion, and yields
²³⁵ $T_2 = (0.053 \pm 0.003)$ ns. Panels **ii** to **iv** in Fig. 5d are visibility traces for resonance fluorescence
²³⁶ for varying excitation powers, as indicated by the Rabi frequencies. It is worth noting that at the
²³⁷ higher powers Rabi oscillations are visible, which are reasonably well reproduced by the model⁵⁵.
²³⁸ The corresponding coherence times are **i** $T_2 = (0.10 \pm 0.1)$ ns, **ii** $T_2 = (0.07 \pm 0.01)$ ns and **iii**
²³⁹ $T_2 = (0.09 \pm 0.01)$ ns, longer than for the non-resonant excitation values. The coherence dynam-

11 - The labeling with "i" is somewhat confusing. If 5d i is non-resonant why is i T2 larger than ii and iii. But you write that i, ii, iii are longer than for the non-resonant values, which should be depicted in 5d i. Maybe it is only me not getting it, but probably you can reduce potential of getting confused.

We thank the reviewer for pointing out the mistake. To make it more clear Figure 5 now has an extra panel with just the non-resonant excitation result (panel e), and panel (d) one with the three resonant excitation ones.

We note, to be sure, that the non-resonant excitation data gives a significantly shorter T2 than all of the resonant excitation traces, as would be expected.

12 - "by > 3 dB/m, than the lowest reported for any foundry-compatible quantum PIC to date." Please check works from QuiX and Xanadu both using SiN chips with ULLW. I am unsure if these companies have lower values but it is worth checking.

We thank the reviewer for this comment. We had indeed researched reported losses from Xanadu and verified that they were of about 0.2 dB/cm (as reported in Nature 591 54 (2021)), being therefore significantly higher than those of ref. 5 in the SI, which employs the same class of waveguide (Si3N4 ridge waveguide). Since the table contains only the lowest reported losses for each waveguide class, we have decided to not include it in our original manuscript. Because we believe this information can be interesting for the general reader, and because of different light source employed, we have included it in the revised version of the manuscript.

We have also realized that the lower losses (0.1 dB/cm) have recently been reported for the double-stripe Si3N4 waveguide, which QuiX employs, than reported on our original table (0.2 dB/cm). In the revised version of the SI, we have added the new reference..

13 - The fabrication and alignment of such an adiabatically tapered structure is very impressive. (Already published by the same authors) It would be great if either a simulation of the impact of misalignment on the coupling efficiency is given in the supplemental or if a reference to a previous publication with the values is included.

We thank the reviewer for the positive comment. We have performed FDTD simulations of the optimized taper, to determine the expected degradation in the modal transformation efficiency as a function of lateral displacement between the GaAs and ULL Si3N4 waveguides. The results, which can be seen in Figure S7(d) of the new supplementary information, indicate that < 90 % degradation (from a maximum of around 0.93) is expected for displacements of up to 400 nm between the two guides. The reasons for such a high tolerance are the large extent of the Si3N4 mode and the optimized phase-matching profile achieved between the two guides.

14 - References: Somaschi, N. et al. Near-optimal single-photon sources in the solid state. There is another paper published slightly earlier: Phys. Rev. Lett. 116, 020401

Thank you for pointing this important reference out, we have included it in our citations list.

15 - Supplemental TABLE S1, missed values from Hyperlight (Loncar) and as mentioned before recent QuiX and Xanadu values. Maybe their current values are published somewhere?

We thank the referee for bringing these up. Indeed, losses reported by Hyperlight and the Loncar group have been in the dB/m range (Zhang et al., Optica 4 1536 (2017)), however at least up to the date of submission, they had not involved quantum photonics demonstrations. Later Loncar group publications, still in the arXiv (Zhu et al., arXiv:2112.09961 (2021)), have demonstrated quantum photonic applications, however on a chip with reported losses of 30 dB/m. Importantly, though, these losses were not explicitly reported in Zhu et al., but rather the

text indicates that the quantum demonstration was performed with the same modulator as in Yu et al. arXiv:2112.09204 (2021), which reported the 30 dB/m value and classical nonlinear optical experiments. We have included both Yu et al. and Zhu et al. on our table.

16 - SiN/InP (nanowire integration) is also missing in the table, probably also with very high losses as far as I can tell.

The reviewer is correct, the SiN/InP nanowire integration work has reported losses of 2.5 dB/cm, and for that reason we had originally chosen to not include it in our table. We have done so in the newer version of the manuscript.

17 - Supplemental RF fitting: Did you consider the coherently scattered laser on top of the Rayleigh peak in the Mollow triplet fitting

The coherently scattered laser and residual pump have been included in the model as a single resolution-limited Lorentzian peak. This statement has been included in the SI text.

Reviewer #3 (Remarks to the Author):

The authors report on the integration of a GaAs nanophotonic device, which embeds InAs quantum dots, on top of a photonic circuit made of Si₃N₄. Together with the characterization of the Si₃N₄ chip, proving low propagation losses, the authors perform quantum dot measurements to prove the injection of single photons in the silicon based circuit. Furthermore, they excite the quantum dot with a resonant laser in the strong driving regime, observing the Mollow triplet spectrum.

The novelty point of this work is based on the low achieved losses in the Si₃N₄ waveguides, while for the integration of the nanophotonic device they employed pick-and-place techniques already proven valid for quantum dots as well as other material platforms.

We thank the reviewer for the concise summary of our work. We would like to emphasize, however, that, while pick-and-place has indeed been demonstrated in prior work (including by our group), application of this method toward ultra-low loss waveguide presents new challenges to the technique, in particular due to the necessity to perform placement into pockets that are deeply etched into a top SiO₂ cladding. As expanded in the main text, a thick upper cladding is a necessary feature of the ultra-low loss waveguide, which supports extremely weakly confined modes. We note furthermore that such weak confinement also creates new challenges for efficient quantum emitter-to-waveguide coupling, which we have addressed in the manuscript. In summary, we believe that the overall challenges faced to allow high coupling efficiency in the present platform - together with the demonstration of waveguide losses that are orders of magnitude better than other pick-and-place demonstrations - significantly differentiate our manuscript from prior work.

Despite the waveguide technology is impressive, I do not think that this work presents enough novelty and impact to be published in Nature Communications. I would rather support the publication in a more specialized journal.

In more details: as said in the abstract, one important goal would be the combination of bright sources with ultra-low loss waveguides. While low loss has been achieved, this value does not play a role in the experiments, since the nanophotonic device is coupled to a straight waveguide (which includes a beamsplitter that is not used in the experiment) instead to a complex photonic chip as shown in Fig.1 (even the spirals are not used in combination with the QD).

We thank the referee for the positive comment regarding the waveguide technology. We respectfully disagree that our work does not present enough novelty and/or impact for Nature Communications. Our reasoning is as follows.

As the reviewer acknowledges, the ultra-low losses achieved in our platform are indeed impressive, being significantly lower than any other so far employed in quantum photonic circuits, and orders of magnitude lower than in any hybrid device demonstrated to date.

While it is true that the low losses do not play a significant role in the reported experiments, our work for the first time shows that such impressively low propagation losses can be achieved in hybrid quantum photonic circuits that incorporate quantum emitter single-photon sources. This is the first report of such performance.

Propagation losses have seldom been reported in hybrid integrated quantum photonic devices. This fact means that a photonic circuit characteristic that is crucial for future scaling and for implementing critical functionalities - such as buffering or time-demultiplexing - is not being addressed with the due attention. Our manuscript goes a long way in correcting such deficiency, in particular in devices that include quantum emitter light sources.

In addition, the QD-to-ULLW coupling is rather modest (below 7%) being limited by design and transfer imperfections. Chanana et al. provide a perspective on how to improve the performances but these approaches may improve some performances at the expense of others, as discussed below.

We point out that the 7 % reported coupling efficiency is not limited by design, but rather primarily by lack of control of the quantum emitter's orientation and position within its hosting nanophotonic geometry. This is discussed in the main text and Section VI of the Supplemental Information. The maximum design collection efficiency, for an emitter that is optimally oriented and positioned in the same geometry as the experimentally demonstrated, would be of ~ 31 %.

This value however could be significantly improved with proper design, as we argue in our response and in Sections VII and VIII of the revised Supplementary Information. Furthermore, as we discuss in the below corresponding comments (#2, #4), our suggested potential improvements to the source efficiency would in no circumstances be detrimental to other features of the platform.

I here provide additional comments which can help strengthening the message for a specialized audience.

1) Figure 1 looks very nice but it is not needed in the paper: it provides a perspective but it does not add useful information for the reader. In particular because, as I understand the paper, the quantum dot is coupled to a straight waveguide, while the spirals are made on another chip and are characterized with a laser (figure 3).

We understand the referee's concern that the figure might be misleading, since the presented work indeed does not provide any experimental results in which quantum emitter single-photons are launched into spirals. However, we respectfully disagree with the opinion that the figure is not needed in the paper. While ultra-low propagation losses are overall crucial for circuit scalability - for reasons that are well established and even relatively easy to grasp by non-specialized readers - Figure 1 illustrates an application that is enabled by ultra-low losses - namely, buffering and time-multiplexing of single-photon qubits. As we point out in the introduction, such capability is essential for a number of current and important photonic quantum computation schemes. We believe that without the figure, this important point will be missed, even by more specialized readers, and for this reason have decided to keep it.

2) The coupling efficiency from the GaAs to the Si₃N₄ is rather modest at this stage (between 4 and 7 %). Apart from the dipole moment orientation, the limited coupling is attributed to misalignment and geometrical imperfections. Therefore, while the Si₃N₄ photonics is of very high quality, the performances are limited by the rest of the processing, so limiting the overall impact.

We thank the reviewer for the careful read of our manuscript. We emphasize that, while the devices experimentally demonstrated in the present manuscript have indeed featured low (< 7 %) coupling efficiency into the ultra-low loss waveguide, the main reason for such low efficiency is the misalignment of the quantum dot's electric dipole transition with respect to the TE GaAs waveguide mode, as is evident on Table S2 of the supplemental information.

We point out, though, that this is not an issue that is inherent or unique to our platform, but rather one that is ubiquitous in photonic devices with functionality enabled by single quantum emitters, if the latter are not deterministically grown or placed within the device. Importantly, in the present work (and in contrast with our work published in Nano Letters 39 7164 (2019)), we have not attempted to create devices with deterministically positioned (and oriented) quantum dots. We therefore argue that the low efficiency observed here is not fundamental, and is absolutely not illustrative of the full potential of our platform.

A different issue, on the other hand, regards the maximum theoretical 31 % extraction efficiency that would be achievable with the fabricated devices, provided the QD can be properly aligned and oriented within the geometry. We argue that such low efficiency is also not representative of the full potential of the platform. Indeed, the specific GaAs geometry that was experimentally tested had not been designed for high coupling efficiency, and had a number of features that altogether contributed to such overall low values.

More specifically, as illustrated in the schematic of Fig. R3_1 below, the geometry launches emitted photons with an efficiency B into a forward traveling straight waveguide mode that is

spatially confined in GaAs ('GaAs mode' in the figure). A mode transformer then converts such GaAs waveguide mode into a Si₃N₄ mode with efficiency η_{MT} , resulting in a total overall QD-to-Si₃N₄ waveguide $\eta_{QD_ULLW} = \beta * \eta_{MT}$. Finite Difference Time-domain (FDTD) simulations of the tested device geometry yielded $\beta = 0.88$, $\eta_{MT} = 0.35$, indicating a major detrimental contribution from the mode transformer.

Fig. R3_1: Schematic of the hybrid single quantum dot single-photon source. Two cross sections are shown, where the relevant features are highlighted. Cross-sectional views of the modes supported in two different sections of the device along z are shown at the bottom. The quantum dot emits into the GaAs mode with efficiency β , which is transformed into the Si₃N₄ mode by the mode transformer, with efficiency η_{MT} . The quantum dot coupling efficiency into the Si₃N₄ waveguide mode is $\eta_{QD_ULLW} = \beta * \eta_{MT}$.

As shown in Section VII of the Supporting Information, however, a more carefully designed mode transformer may be implemented to yield $\eta_{MT} = 0.92$, which would readily provide an overall efficiency $\eta > 0.8$. Further improvement could be achieved in a variety of ways, for instance by increasing β . We refer the referee to our recent work placed in the arXiv (Melo et

al., arXiv:2206.01043), which describes geometries providing $\beta > 0.92$ with Purcell factors of > 20 , in geometries in which the quantum dot is located > 300 nm away from any etched sidewalls. Such a design is completely compatible with the present platform, and has the potential to circumvent, or at least improve, via both the Purcell enhancement and distancing from etched sidewalls, parameters such as single-photon indistinguishability, spectral diffusion and blinking. To illustrate the compatibility of such geometries with our platform, Fig R3_2 below shows a GaAs geometry embedded in SiO₂ that was designed with the same inverse-design approach as in the above arXiv manuscript. The simulated geometry yields $b=0.91$ into the GaAs output waveguide mode, with a Purcell factor of 10. This design was included as Section VIII in the revised SI.

Fig. R3_2: Example GaAs nanophotonic geometry, embedded in SiO₂, that provides $b=0.91$ into the output GaAs waveguide mode, same as in Fig. R1, with a Purcell radiative enhancement factor of 10. As in Fig. R1, light from such GaAs mode can be converted into the Si₃N₄ mode with the optimized mode transformer discussed in the SI, yielding an overall efficiency $\eta \sim 0.85$. The top figure shows the refractive index distribution of the device, whereas the bottom shows the electric field distribution for an optimally positioned quantum dot.

Further improvements could potentially be achieved by employing a high quality factor cavity evanescently coupled directly to the Si₃N₄ waveguide. As shown in Katsumi et al. *Optica* 6 786 (2019), $\eta > 99\%$ can potentially be achieved, albeit over a relatively narrow wavelength band.

3) Following the previous point, if a different dipole orientation would increase the coupling up to 30% (as simulated in the SI), the question arises on the fabrication reproducibility. In other words, how many samples have been fabricated? Statistically, some of them would have the correct dipole orientation increasing the performance.

Due to a number of issues in device fabrication associated primarily with the pick and place procedure, only three full GaAs devices were successfully transferred onto Si₃N₄ waveguides. The data shown in the manuscript was taken from the device that presented the highest outcoupling efficiency. The chances of finding one optimally positioned and oriented dot within such a small sample would be very low, though we argue that even within a considerably larger sample such a feat would be highly unlikely.

It is worth noting also that our pick-and-place process has not been optimized for transferring the relatively large and pliable GaAs devices, and for this reason our yield was low. Besides optimization of the particular setup and technique, a couple of alternatives could be used to improve the transfer yield. For instance, using arrays of emitters on a single chiplet, as demonstrated in Wan et al., *Nature* 583, 226–231 (2020) would potentially allow multiple sources to be transferred at once. Another possibility would be to perform the GaAs device layer transfer prior to the ULLW top SiO₂ cladding deposition, either via transfer-printing [Katsumi et al., *Optica* (2019)] or wafer bonding [Davanco et al., *Nature Comms* (2017)]. Naturally, all alternatives would only solve the photonic hybridization yield issue, not the low yield associated with lack of dot positioning control.

4) In paragraph II, it is said that a mode transformer needs a small spacing (between transformer and waveguide), which has not been used in the current geometry. Can the achieved low propagation losses be maintained even modifying the WG geometry to achieve an increased in-coupling? Since it is said that it can be done to reach 93%, the question may arise why it has not been done, getting an efficiency of 7%.

We thank the referee for the careful reading of our manuscript. We believe that the description of our device at the beginning of Section II had some unnecessary information that, though correct, created some confusion.

What we tried to express was the fact that, generally, a small spacing t , with $0 \leq t \leq 250$ nm, is needed between the GaAs and Si₃N₄ waveguides, to ensure evanescent coupling between the two. In the case of our device, that spacing was zero, so the GaAs sat directly on top of the Si₃N₄ waveguide core. This was accomplished by etching a pocket onto the ULL waveguide's top SiO₂ cladding and placing the GaAs device in the pocket, then covering the pocket with SiO₂. This is all indicated in Fig. 2.

The 93 % mode transformer coupling efficiency predicted with the optimized geometry discussed in the Supplementary Information was indeed achieved in modeling by considering zero spacing between the GaAs and Si₃N₄ guides and a top SiO₂ clad everywhere, exactly as

shown in Fig. 2. The zero spacing does not affect the losses in the ULL waveguide lengths that are away from the GaAs portions, since in such regions (the so-called *passive* circuit sections) the top SiO₂ cladding, and hence the ULLW, is fully preserved. We re-emphasize the fact that the ULLWs are nowhere affected by the integration of the GaAs sources except where the two come into close proximity. This only takes place inside the pockets, over lengths of a couple of tens of microns, and overall losses are insignificant over such short lengths.

In short, we believe that the referee's concern that optimized coupling structures may lead to higher losses in the passive ULL circuit portions is unjustified.

On the other hand, we emphasize that the 93 % value that the referee mentions corresponds only to the efficiency of an optimized mode transformer, not the overall single-photon coupling efficiency into the Si₃N₄ waveguide. This is discussed more at length in our response to comment 2). We believe that some of the referee's confusion regarding the achieved and achievable coupling efficiencies is due to the fact that such a discussion occurred prematurely in the manuscript. Indeed, as the referee points out, the optimized mode transformer was not experimentally demonstrated, rather one with sub-optimal theoretical efficiency was so. To avoid being misleading, we have moved the main text discussion about the optimized mode transformer design to the discussion in Section IV.

We have made changes to the second paragraph of Section II to clarify these points. The second paragraph of Section II now reads:

“To ensure evanescent coupling between the GaAs and Si₃N₄ layers using the mode transformer, the QD-containing GaAs device is placed in direct contact with the top of the Si₃N₄ guide. This is accomplished by first etching a pocket into the 1 μm top SiO₂ upper cladding of the ULLW, down to its SiN core, and then placing the GaAs device into the pocket, as seen in Fig.2a. In a following step, the placed GaAs device is covered with a 1 μm thick SiO₂ cladding layer, as shown in Fig. 2b and c. It is with noting that portions of the Si₃N₄ ULL that are distant from the placed GaAs device are completely unaffected by our processing, since the top SiO₂ cladding is preserved everywhere. Finite Difference Time-domain (FDTD) simulations predicted that the fabricated geometries could yield a maximum theoretical single-photon coupling efficiency ~ 0.31 into the Si₃N₄ waveguide. Sections IV in the main text and VII and VIII in the SI discuss concrete alternative geometries that have the potential to achieve $\eta > 0.8$ ”

We have moved the paragraph regarding the optimized mode transformer geometry from Section II into Section IV, and expanded the discussion to accommodate further comments from Reviewers 1 and 3.

5) In paragraph III, linear losses for the spirals were found. Why losses for the 3 meter spiral are almost three times smaller than for the 2 meter one (in terms of dB per meter)? Is somehow the fabrication more effective for long spirals or is it related to low sample statistics?

We note that the 2.8 dB/m losses measured for the 2 m spiral corresponds only to a 34 % (~ 1.8 dB increase). The higher losses are more likely due to higher overall waveguide bending losses.

As discussed in Bauters, et al. Optics Express 31 544 (2013) (ref. [32] in the revised manuscript), bending losses can be reasonably well modeled as $a_2 = a_1 \exp(-a_2 r(z))$, where $r(z)$ is the local radius of curvature (RoC) along the spiral. The smaller overall ROCs for the shorter spirals are then expected to lead to higher propagation losses.

6) In paragraph IV, the used quantum dot is described but some important information is missing: which emitter transitions are X1, X2 and X3? In particular X1, is it a neutral exciton, a charged exciton or a biexciton?

We thank the reviewer for the comment. We have created Section IV in the Supplementary Information, that contains a characterization of the radiative lifetime, power series and second-order cross-correlation of the three transitions. Based on this new data, we have relabeled transitions X1, X2 and X3 as X0, C1 and C2. Line X1 (now called X0) presented a power series slope of close to 1, so we believe it to be the quantum dot's neutral exciton. Cross-correlation measurements between the additional lines did not reveal the asymmetric bunching characteristic of biexciton cascades, so we have assigned the two lines to charged exciton transitions and labeled them C1 and C2.

7) Again in paragraph IV, the quantum dot photons are sent through a MMI beamsplitter but it seems that for the Hanbury-Brown and Twiss setup an external beamsplitter is employed. This would mean that the MMI is only an additional source of losses (around 45% as in the SI). As said before, the experiment does not really make use of the low loss Si₃N₄ for performing the measurements.

The presence of the MMI was simply to allow passive characterization for (potential) future work, it was never intended as part of a more significant demonstration for the present work. Indeed, HBT measurements using on-chip waveguides have been demonstrated many times, so the value of such an experiment, which would be challenging considering the need for endfire coupling with a pair of optical fibers (as opposed to vertical grating coupling as done elsewhere) was not apparent at the time of the experiment.

However, In the SI we provide QD spectra obtained from the two MMI arms, indicating that the beamsplitting operation works as expected, with a split-ratio of roughly 50 %, which would be sufficient to deduce the MMI's contribution to an HBT measurement with ultra-low-loss circuit.

8) The authors refer in the text to Figure 1 a or b when they mean figure 2. In the methods section, they refer to figure 4e (it should be d) and fig. 4b (should be a).

We thank the reviewer for noticing these mistakes. We have made changes accordingly.

9) In figure 4 (and describing text) a $g(2)$ of 7% is reported. Since the filter used has 500 pm bandwidth (as written in the SI), the $g(2)$ is still good, despite the underlying background of X1 within 500pm range. Has the background being subtracted? It can be helpful to mention it.

We thank the reviewer for the suggestion. Indeed, the background has not been subtracted. We have changed the text accordingly.

10) The T1 value reported for resonant excitation has been obtained by fitting the $g(2)$ in figure 5. It would be useful to report which excitation power has been used and if deconvolution has been performed in the fit in fig.5b.

We thank the reviewer for the comment. We have added the information to the manuscript.

11) Regarding figure 5d, the authors should comment on why using a Gaussian fit instead of a Voigt (which would include the Lorentzian component from the lifetime).

The fitting was in fact done with a weighted sum of a Gaussian and a two-sided exponential, as indicated in eq.(10) in Section X of the SI. We have corrected the associated text in the manuscript. In addition we have provided the fit amplitudes for Gaussian and exponential components for the MZI visibility traces on Table S3 in Supplementary Information Section X. For the p-shell data alluded to by the referee, the two components have indeed comparable amplitudes.

12) In paragraph VI, approaches to achieve better performances are discussed, but it is not clear if these improvements will come at the expense of other parameters. For example, increasing the in-coupling may need different waveguide geometries. How the losses will be impacted?

We thank the reviewer for bringing up this important point. All improvements that were suggested towards enhancing the single-photon coupling efficiency into the ultra-low loss Si₃N₄ waveguides and single-photon indistinguishability regarded only the GaAs layer of our platform. These suggested changes would minimally affect both propagation and coupling losses. Specifically:

1 - None of the suggested changes will impact the losses in the purely passive portions of the circuit where the GaAs is not present. As we point out in our introduction, and as motivated by Fig. 1 and device schematics in Fig. 2, the III-V devices are to be introduced only in small portions of the circuit, whereas the passive portions, which would e.g. implement linear optical elements, buffers and delays, would remain untouched.

2 - Losses at the transitions between the GaAs and Si₃N₄ waveguides would also not be affected, provided the quantum dot couples efficiently to the fundamental GaAs TE mode ('GaAs mode' in Fig. R3_1 above) that the current mode transformers were designed for.

3- Efficient coupling of quantum dots single-photons to the fundamental GaAs waveguide TE mode can potentially be affected by some of the suggested solutions. For instance, the need to create geometries with etched sidewalls that are sufficiently distant (> 300 nm) from the quantum dot creates challenges, because the necessary dimensions are consistent with multimode waveguide propagation. Such a feature increases the likelihood of non-exclusive quantum dot coupling to high-order GaAs waveguide modes, which would eventually result in lower coupling efficiencies into the ultra-low loss Si₃N₄ mode. While we believe a number of integrated photonic techniques can be used to circumvent such a problem, we refer the referee to our recent work on the inverse-design of quantum emitter single-photon sources

(<https://arxiv.org/abs/2206.01043>), which presents a nanophotonic source geometry that addresses the outcoupling challenge described above, while simultaneously providing a high Purcell factor that may be leveraged to enhance single-photon indistinguishability. This is a geometry that could be implemented within our platform (an example is given in Fig. R3_2 and related discussion above).

4 - Generally, our platform allows the creation of high resolution nanophotonic geometries in the GaAs layer, and that imbues it with great flexibility to overcome limitations that might be introduced depending on the type of problem that needs to be solved. For instance, if waveguide-based resonant excitation is desired, the multimode waveguide scheme presented in Uppu et al., Nat. Comms 11 5198 could be implemented, as described in Fig. R1_3 of our response to Referee #1.

All of the above changes would be implemented in the GaAs layer alone, without affecting the low-loss Si₃N₄ waveguide portions of the circuit; and all emitted photons would still be efficiently launched into the ULL TE modes of the Si₃N₄ with the same optimized mode transformer as designed in the SI. Therefore, concerns that ultra-low losses may be impacted by such changes are also unjustified.

In the revised manuscript, we comment on this fact in Section II.

13) The discussion in paragraph VI on polarization entangled photons from a quantum dot is a bit misleading. To my understanding, the used waveguides support only one polarization, which would impact the polarization entanglement. Using waveguides which support both polarizations may impact the losses (which are the main novelty point of the work).

We thank the reviewer for the comment, we agree that without proper context the discussion might indeed be misleading. Succinctly, a different type of geometry would have to be employed in this case, since the single TE mode waveguide that has been experimentally demonstrated.

While many possibilities exist, the multimode GaAs waveguide that was investigated in Jin et al., Nano Lett. 22 586 (2022) offers one potential solution. In that work, polarization-entangled photon pairs from a single quantum dot were coupled to two TE modes (the fundamental and a third-order one) of the same multimode GaAs waveguide, so coupling of the two photons was to the same polarization. Such a geometry could be easily produced in our platform, since it consists of simply a slightly wider waveguide than ours, and the presence of a SiO₂ / Si₃N₄ substrate would not functionally affect it.

Photons coupled to the high-order mode can then be extracted into the fundamental TE mode of a separate waveguide via an intermodal directional coupler (see e.g. Mohanty et al., Nature Comms. 8 14010 (2017)). The same mode transformer geometry as demonstrated here may then be employed to convert the TE GaAs modes into the ultra-low loss nitride waveguide.

We note that more efficient alternative geometries are the subject of a current investigation in our group, and will be reported in the near future.

All of the above changes would be implemented in the GaAs layer alone, without affecting the low-loss Si₃N₄ waveguide portions of the circuit; and all emitted photons would still be efficiently launched into the ULL TE modes of the Si₃N₄ with the same optimized mode transformer as designed in the SI. Therefore, concerns that ultra-low losses may be impacted by such changes are also unjustified.

To address the referee's comment in the main text, the following phrase has been edited in the Discussion section:

For instance, polarization-entangled photon pairs may be generated from the biexciton-exciton cascade [Liu et al., Nature Nanotech 2019], where the two states are typically split by ~ 1 nm. These entangled photon states, when captured into an integrated photonic circuit - for instance via two TE modes of a multimode GaAs waveguide [Jin et al., Nano letters 2022] - could present interesting opportunities for quantum information processing on a chip. Importantly, all of the suggested options for improving the source efficiency would only involve modifications to the GaAs device layer, whereas the Si₃N₄ ultra-low loss portions of the circuit would remain unaffected.

We note lastly that the issue of supporting only TE modes in ULL WGs and operating as a TM mode filter is discussed in Bauters, J. F., et al. (2013). "Ultralow-Loss Planar Si₃N₄ Waveguide Polarizers." IEEE Photonics Journal 5(1): 6600207-6600207. Distinguishing between modes by introducing higher loss for one mode than the other can be achieved by adjusting the bending radius such that TM bend losses are much higher than TE. Therefore it depends on the type of structure that one is coupling to in order for there to be a mode filter. This is not a feature of the high aspect ratio waveguide by itself. This is an important issue to study in future work and is outside the scope of the current manuscript, since we report coupling only to a straight waveguide even though afterwards the light can circulate in a spiral.

In conclusion, I appreciate the technological steps forward made by the authors but this study should be published in a more technical journal. The proposed changes may provide additional interesting information for a specialized audience.

Once again, we thank the referee for the constructive feedback and hope that our changes are not only useful for a more specialized audience, but also address why the manuscript is suitable for Nature Communications.

Major changes to the main text and supplementary information:

1 - To address referee #1's comments 1 and 2, we have added data comparing quantum dot spectral diffusion under non-resonant and resonant excitation. Data and a discussion are provided in **Section XII of the new Supplementary Information**, as well as **Table S3 in Section X of the Supplementary Information**.

2 - To address referee #1 and #2's general comments about single-photon coupling efficiencies, we have made revisions to the **Device Description and Fabrication Section** of the **main text**, to better describe the functioning of the tested devices, and to clarify how significant

improvement may be achieved through better photonic device. **Section VII** in the **Supplementary Information** was created to further support the latter claim.

3 - To address referee #2's comment 3 , we created **Section XI of the Supplementary Information**, with data showing triggered single-photon generation under resonant excitation.

4 - Based on referee #2 and #3's comments 8 and 6 respectively, added data regarding identification of quantum dot emission lines. The new data and discussion are in **Section IV** of the revised **Supplementary Information**.

5 - Edited **table S1** to include work referred to by Referee #2.

6 - Edited the first paragraph of the **Resonance Fluorescence Section** of the main text, to address referee #2's comment 10.

REVIEWERS' COMMENTS

Reviewer #1 (Remarks to the Author):

The authors addressed all of the raised questions, and I have no other comments.

Reviewer #2 (Remarks to the Author):

The revised manuscript of Ashish Chanana et al. has addressed all my comments and added several important aspects in the supplemental material. Thus I recommend publication in Nature Communication. I only suggest to either remove the non-resonant lifetime measurement or to rewrite this part, since the resonant lifetime measurement accurately gives the lifetime of the excited state, whereas the non-resonant measurement also includes different relaxation processes and the extracted decay times are very hard to interpret.

In addition, while reading the discussion with reviewer 1 on RF on-chip, the authors might want to have a look at this recent paper: Nature Communications 13, 3982 (2022).

Reviewer #3 (Remarks to the Author):

I would like to start by thanking the authors for their efforts in revising their manuscript and add more measurements and information, in particular regarding the quantum dot.

Achieving low loss waveguides, which are 50% better (factor of 5 for Si₃N₄) than previously reported systems is indeed a nice result. Also proving that the authors' developed pick-and-place can be used to combine III-V structures with low loss waveguides represents a nice step forward.

That being said, I still believe that the results, despite impressive, represent more a technological step forward rather than a scientific advance. The low losses observed in the waveguide play no role in the experiments: a breakthrough result would be showing measurements that can only be done because of the high quality waveguides.

In addition:

1) As said in the authors' response, realizing the hybrid integration of III-V on Si₃N₄ is more challenging than before due to deeply etched pockets: it is important that the authors solved this issue, but its nature is rather technological than scientific.

2) I really appreciate the detailed theoretical analysis of the current performances limitations and perspective on how to improve them. Nonetheless, they still have to be implemented. It is true that in the current geometry a maximum coupling efficiency of 31% can be achieved, but due to dipole alignment this value is now pinned to 7% (as said, other 2 devices showed even lower coupling). To my understanding, this number is the most critical one when it comes to scalable quantum photonics.

For these reasons, I would rather recommend publication in a more specialized journal. The technological advancements are impressive and deserve publication but they do not enable, here, special measurements that could not be done otherwise.

We thank the reviewers for their final comments on our manuscript. Below we have point-by-point replies to all of the points that have been brought up. We hope that our replies are sufficient to warrant publication in Nature Communications.

The original referees' comments are in printed black, whereas our responses are in blue.

REVIEWERS' COMMENTS

Reviewer #1 (Remarks to the Author):

The authors addressed all of the raised questions, and I have no other comments.

Our reply: We thank the reviewer for all the feedback, and for ultimately agreeing with publication of our manuscript in Nature Communications.

Reviewer #2 (Remarks to the Author):

The revised manuscript of Ashish Chanana et al. has addressed all my comments and added several important aspects in the supplemental material. Thus I recommend publication in Nature Communication. I only suggest to either remove the non-resonant lifetime measurement or to rewrite this part, since the resonant lifetime measurement accurately gives the lifetime of the excited state, whereas the non-resonant measurement also includes different relaxation processes and the extracted decay times are very hard to interpret.

In addition, while reading the discussion with reviewer 1 on RF on-chip, the authors might want to have a look at this recent paper: Nature Communications 13,: 3982 (2022).

Our reply: We thank the referee for recommending publication of our paper, and for the helpful comment regarding the non-resonant pump lifetime measurement. We have decided to remove that measurement from the main text, seeing that it does not add significantly to the discussed results. We have also cited Nature Communications 13,: 3982 (2022).

Reviewer #3 (Remarks to the Author):

I would like to start by thanking the authors for their efforts in revising their manuscript and add more measurements and information, in particular regarding the quantum dot.

Achieving low loss waveguides, which are 50% better (factor of 5 for Si₃N₄) than previously reported systems is indeed a nice result. Also proving that the authors' developed pick-and-place can be used to combine III-V structures with low loss waveguides represents a nice step forward.

Our reply: We thank the reviewer for the positive comments.

That being said, I still believe that the results, despite impressive, represent more a technological step forward rather than a scientific advance. The low losses observed in the waveguide play no role in the experiments: a breakthrough result would be showing measurements that can only be done because of the high quality waveguides.

Our reply: It is widely accepted that low loss is paramount in QIP, and in many ways the main issue. As we argue in our manuscript, ultra-low losses integrated photonic waveguides can be central to a wide class of quantum computation systems, and quantum emitter single-photon sources can provide the necessary photon flux and close-to-deterministic operation necessary for photonic systems of scale. Demonstrating that single-photon sources and ultra-low loss waveguides can be brought together is therefore a milestone result in itself. It is worth noting also that ultra-low loss waveguides had not been applied to a QIP experiment before.

In addition:

1) As said in the authors' response, realizing the hybrid integration of III-V on Si₃N₄ is more challenging than before due to deeply etched pockets: it is important that the authors solved this issue, but its nature is rather technological than scientific.

The ability to bring together the two types of photonic structures is a problem that in our view involves more than a new fabrication process. As we argue in our manuscript, due to the nature of the ultra-low loss waveguide platform, photonic design for quantum dot to ULL waveguide coupling is a challenging and important aspect, which in our view has a scientific, if applied, character. In particular, inverse design (in particular electromagnetic design) is an important, current scientific research topic that shows great promise for solving problems, such as creating efficient photonic quantum emitter interfaces, which, given all the constraints and necessary trade-offs required, are very challenging to tackle through conventional methods.

2) I really appreciate the detailed theoretical analysis of the current performances limitations and perspective on how to improve them. Nonetheless, they still have to be implemented. It is true that in the current geometry a maximum coupling efficiency of 31% can be achieved, but due to dipole alignment this value is now pinned to 7% (as said, other 2 devices showed even lower coupling). To my understanding, this number is the most critical one when it comes to scalable quantum photonics.

The problem of deterministic quantum dot (or, more generally, quantum emitter) positioning and orientation within a photonic structure affects the vast majority of single quantum emitter work, goes somewhat beyond the scope of the present work. Our reasoning is as follows. While we agree that direct experimental demonstrations as suggested by the referee would be preferable, we believe our work shows a number of ways in which high coupling efficiencies can be achieved in our platform. Without such measures, poor coupling efficiencies are to be expected independently of the dipole orientation. Conversely, provided the capability for positioning and alignment of QDs is available, high coupling efficiencies would be achievable. This is a topic of future work in our group and in a number of groups working on single quantum emitter devices.